# IL-6/STAT3 pathway induced deficiency of RFX1 contributes to Th17-dependent autoimmune diseases via epigenetic regulation

Ming Zhao[1], Yixin Tan[1], Qiao Peng[1], Cancan Huang[1], Yu Guo[1], Gongping Liang[1], Bochen Zhu[1], Yi Huang[1], Aiyun Liu[1], Zijun Wang[1], Mengying Li[1], Xiaofei Gao[1], Ruifang Wu[1], Haijing Wu[1], Hai Long[1] & Qianjin Lu[1]

Epigenetic modifications affect the differentiation of T cell subsets and the pathogenesis of autoimmune diseases, but many mechanisms of epigenetic regulation of T cell differentiation are unclear. Here we show reduced expression of the transcription factor RFX1 in CD4$^+$ T cells from patients with systemic lupus erythematosus, which leads to IL-17A overexpression through increased histone H3 acetylation and decreased DNA methylation and H3K9 tri-methylation. Conditional deletion of *Rfx1* in mice exacerbates experimental autoimmune encephalomyelitis and pristane-induced lupus-like syndrome and increases induction of Th17 cells. In vitro, Rfx1 deficiency increases the differentiation of naive CD4$^+$ T cells into Th17 cells, but this effect can be reversed by forced expression of Rfx1. Importantly, RFX1 functions downstream of STAT3 and phosphorylated STAT3 can inhibit RFX1 expression, highlighting a non-canonical pathway that regulates differentiation of Th17 cells. Collectively, our findings identify a unique role for RFX1 in Th17-related autoimmune diseases.

[1] Department of Dermatology, The Second Xiangya Hospital of Central South University, Hunan Key Laboratory of Medical Epigenomics, Changsha, Hunan 410011, China. Ming Zhao, Yixin Tan, Qiao Peng and Cancan Huang contributed equally to this work. Correspondence and requests for materials should be addressed to M.Z. (email: zhaoming307@csu.edu.cn) or to Q.L. (email: qianlu5860@csu.edu.cn)

Systemic lupus erythematosus (SLE) is a complex auto-immune disorder resulting in multi-organ destruction[1]. Autoantibody deposition and inflammatory cell infiltration in target organs such as kidneys and brain cause severe complications. Interleukin (IL)-17-producing CD4[+] type 17 helper T (Th17) cells are defined by specific developmental and functional features that are distinct from those of Th1 and Th2 cells[2,3]. Th17 cells primarily produce two members of the IL-17 family, IL-17A and IL-17F, which promote local chemokine production to recruit monocytes and neutrophils to sites of inflammation[4]. Aberrant Th17 cell differentiation, linked to increased Th17 cytokine production, is an important contributor to SLE pathogenesis[5].

Epigenetic factors have a pivotal function in the pathogenesis of SLE[1]. DNA hypomethylation, increases in histone acetylation, and decreases in histone 3 lysine 9 (H3K9) and lysine 27 (H3K27) contribute to the overexpression of several autoimmune-related genes, such as CD70 and CD11a, which leads to the auto-reactivation of T cells in SLE[6–10]. Epigenetic modifications are mediated by transcription factors in T cell subset differentiation[11–13]; the transcription factor Etv5 recruits histone-modifying enzymes to the *Il17a-Il17f* locus, resulting in increased active histone marks and decreased repressive histone marks, thereby promoting IL-17A and IL-17F expression and Th17 differentiation[14]. Also, cAMP-responsive element modulator α (CREMα) protein induces IL-17A expression and mediates histone acetylation and DNA methylation at the *IL17A* gene locus through the recruitment of histone deacetylase 1 (HDAC1) and DNA methyltransferase 3a (DNMT3a) in patients with SLE[15].

Our previous studies identified a decrease in transcription factor RFX1 expression in CD4[+] T cells from SLE patients[16]. RFX1 is a member of the regulatory factor X (RFX) family of transcription factors, which can bind X-box of the DNA sequence to regulate the expression of target genes[17]. RFX1 has a C-terminal repressive region, an overlapping dimerization domain, and an N-terminal activation domain and is able to activate or repress target gene transcription depending on the cellular context[18]. RFX1 has an important function in brain tumors and sensorineural hearing loss[19,20]. However, its function in immune cells and the immune response is unclear. Our previous studies showed that RFX1 regulates the immune-related genes CD11a and CD70 by affecting the epigenetic modifications of the two genes' loci in CD4[+] T cells and that downregulation of RFX1 in CD4[+] T cells contributes to the autoimmune response of SLE patients[16,21].

In this study, we investigate the function of RFX1 in aberrant Th17 differentiation in SLE patients. Our results indicate that RFX1 can repress IL-17A gene expression, and its deficiency contributes to increased IL-17 production and Th17 differentiation in patients with SLE. Rfx1 conditional deletion in mice exacerbates experimental autoimmune encephalomyelitis (EAE) and pristane-induced lupus-like mice models. We also show that downregulation of RFX1 causes increased histone acetylation and decreased histone 3 lysine 9 (H3K9) tri-methylation and DNA demethylation in the *IL17A* locus, thereby leading to over-expression of IL-17A and increasing Th17 cell differentiation. In addition, RFX1 functions downstream of signal transducer and activator of transcription factor 3 (STAT3) and IL-6-induced STAT3 activation can reduce the expression of RFX1. These data support a role for RFX1 deficiency in the pathogenesis of SLE, by mediating aberrant Th17 differentiation induced by IL-6 in the inflammatory environment.

## Results

**RFX1 expression is decreased in SLE CD4[+] T cells.** Our pre-vious studies have shown that the expression and activity levels of RFX1 are decreased in CD4[+] T cells from patients with SLE compared with healthy controls[16]. To gain a better understanding of the role that RFX1 plays in Th17 cell differentiation, we measured the mRNA and protein expression levels of RFX1 and IL-17A in CD4[+] T cells from a new cohort of SLE patients, including 23 active patients and 8 inactive patients, and 14 healthy controls. The mRNA and protein expression levels of RFX1 were significantly decreased in CD4[+] T cells of inactive and active SLE patients compared with healthy controls (Fig. 1a–c). By contrast, both IL-17A mRNA expression levels in CD4[+] T cells and IL-17A protein levels in serum were significantly elevated in SLE patients compared with healthy controls (Fig. 1d, e). Fur-thermore, we found that the mRNA and protein levels of RFX1 were inversely correlated with the mRNA and serum protein levels of IL-17A (Fig. 1f, g). In addition, we also measured the expression of other genes associated with Th17 cells; however, there were no significant changes in mRNA expression levels of IL-17F and RORC between SLE CD4[+] T cells and healthy con-trols (Fig. 1h, i).

**RFX1 inhibits IL-17A expression in CD4[+] T cells.** To determine whether RFX1 regulates IL-17A expression in CD4[+] T cells, we isolated CD4[+] T cells from healthy controls and transfected them with RFX1 small interfering RNAs (siRNAs) and negative con-trol, along with stimulation from anti-CD3 and CD28 antibodies. RFX1 protein levels were decreased in CD4[+] T cells transfected with two siRNAs compared with negative control siRNA. IL-17A mRNA and protein expression levels were significantly increased in CD4[+] T cells with RFX1 knockdown compared with negative control (Fig. 2a–c). By contrast, we overexpressed exogenous RFX1 in SLE CD4[+] T cells through transfection of an RFX1 expression plasmid. When RFX1 expression was increased in SLE CD4[+] T cells, IL-17A mRNA and protein expression levels were significantly decreased compared with empty vector control (Fig. 2d–f). Altogether, these data show that RFX1 represses IL-17A expression, and RFX1 loss can lead to IL-17A over-production in CD4[+] T cells.

**RFX1 regulates chromatin modifications in the *IL17A* locus.** To investigate the mechanism of RFX1 regulation of IL-17A expression, we predicted the putative binding sites of RFX1 in the promoter region of the *IL17A* gene. According to the predictions of the Matinspector and TFBIND software[22,23], two binding sites for RFX1 were found in the ~468–452 bp (Site 1) and ~639–623 bp (Site 2) regions upstream of the transcription start site (TSS) of *IL17A* (Fig. 3a). To investigate whether RFX1 regulates *IL17A* expression at the transcriptional level, we performed luciferase assays using reporter constructs spanning Site 1 and Site 2 of the human *IL17A* promoter. Indeed, RFX1 overexpression resulted in significant downregulation of *IL17A* promoter activities in the wild-type (WT) reporter construct. Next, we mutated Site 1 and Site 2, both separately and together, in the WT *IL17A* reporter construct and noted that the promoter activity was still repressed with both the individual Site 1 and Site 2 mutant reporter con-structs. When both sites were mutated, RFX1 did not alter the promoter activity. These results suggest that both sites are crucial for RFX1-mediated *IL17A* transcription in human CD4[+] T cells (Fig. 3b). Moreover, we used biotin-tagged specific probes for Site 1 or Site 2 to perform an electrophoretic mobility shift assay (EMSA) with an RFX1-specific antibody. The results showed that RFX1 could bind to Site 1 and Site 2 in the *IL17A* promoter (Fig. 3c). We also addressed the in vivo relevance of RFX1 binding to the *IL17A* promoter by chromatin immunoprecipita-tion (ChIP) assays. We observed marked RFX1 binding in the

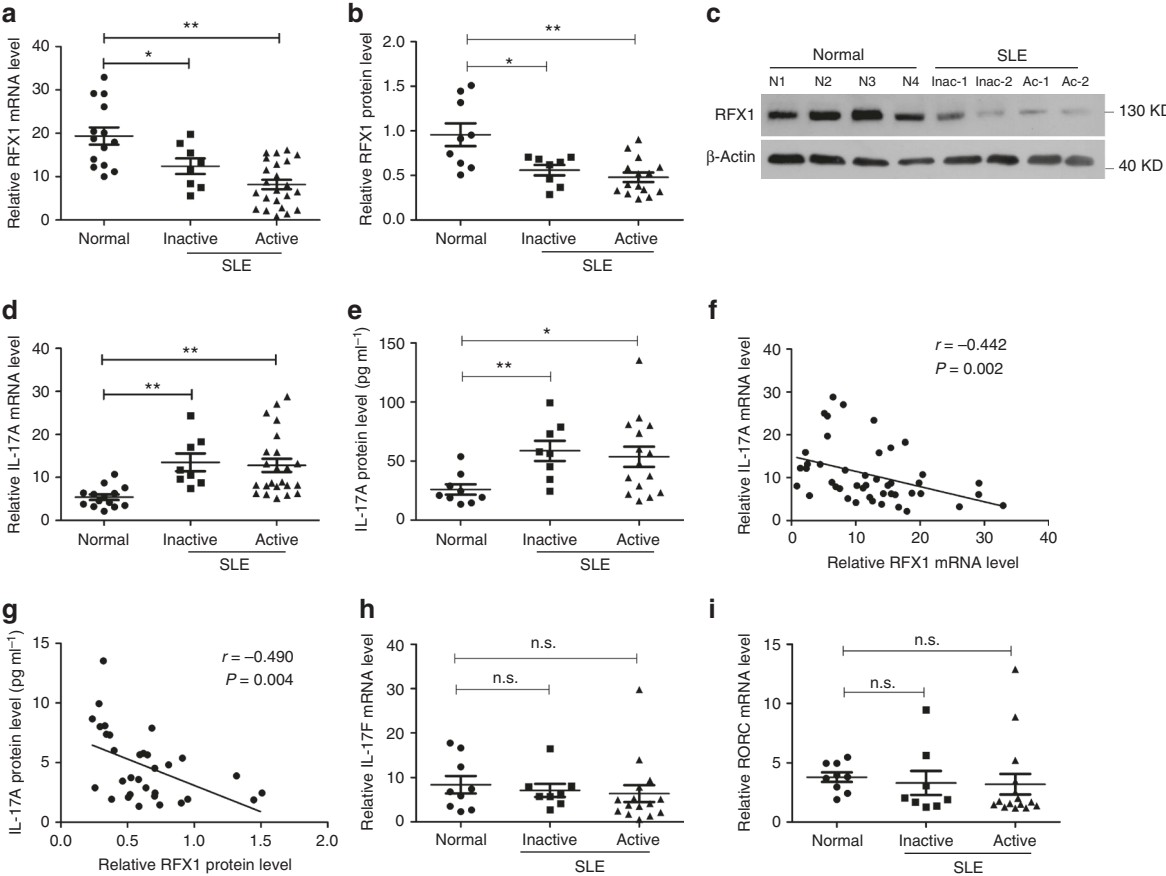

**Fig. 1** Decreased RFX1 and increased IL-17A expression in CD4$^+$ T cells of SLE patients. mRNA (**a**) and protein (**b**) expression levels of RFX1 in CD4$^+$ T cells of inactive SLE patients ($n = 8$), active SLE patients ($n = 23$ for mRNA and $n = 15$ for protein) and healthy subjects ($n = 14$ for mRNA and $n = 9$ for protein). Representative results of western blot analysis are shown in (**c**). IL-17A mRNA expression level in CD4$^+$ T cells (**d**) and protein level in serum (**e**) of inactive and active SLE patients and healthy subjects. **f** Correlation between RFX1 and IL-17A at the mRNA level. **g** Correlation between RFX1 protein levels in CD4$^+$ T cells and IL-17A protein levels in serum. **h** and **i** mRNA expression levels of IL-17F and RORC in CD4$^+$ T cells of inactive SLE patients ($n = 8$), active SLE patients ($n = 15$) and healthy subjects ($n = 9$). Small horizontal lines indicate the mean ($\pm$s.e.m) in **a**, **b**, **d**, **e**, **h**, **i**. *$P < 0.05$; **$P < 0.01$; n.s., not significant, compared between the indicated groups. Student's $t$-test (two-tailed) and Mann–Whitney $U$-test (two-tailed) were used to compare the results. Sperman's correlation coefficient was used for the correlation analysis (two-tailed)

region containing Site 1 and Site 2 in the *IL17A* promoter in CD4$^+$ T cells of healthy subjects (Fig. 3d).

Our previous studies have shown that RFX1 regulates levels of DNA methylation, H3 acetylation (H3ac), and H3K9 tri-methylation (H3K9me3) in the promoters of *CD11a* and *CD70* in SLE CD4$^+$ T cells[16,21]. DNA methylation of the *IL17A* promoter has been shown to be decreased in CD4$^+$ T cells of SLE patients compared with healthy controls[15]. To characterize RFX1 regulation of histone modifications in the *IL17A* promoter, we first compared the RFX1 binding level and the H3ac and H3K9me3 levels between SLE CD4$^+$ T cells and healthy controls by ChIP-quantitative PCR (qPCR). Two primers spanning RFX1binding Site 1 or Site 2 and one distal primer were used to amplify the ChIP product (Supplementary Fig. 1a). The results showed that the H3ac level was higher and that the H3K9 me3 level and the RFX1-binding level were lower in CD4$^+$ T cells of SLE patients compared with healthy controls, which suggests that the changed histone markers may be related to RFX1 binding in the *IL17A* promoter region in SLE CD4$^+$ T cells (Supplementary Fig. 1b–d).

Furthermore, we measured the changes in H3ac, H3K9me3, and DNA methylation levels in the promoter of *IL17A* in CD4$^+$ T cells transfected with RFX1-siRNA or an RFX1 expression plasmid. We found that the H3ac level was elevated while the H3K9me3 level and DNA methylation levels of several CG pairs within the *IL17A*

promoter region were downregulated in normal CD4$^+$ T cells transfected with RFX1-targeting siRNAs compared with negative control (Fig. 4a–c). By contrast, we found that the H3ac level was downregulated and H3K9me3 level and DNA methylation levels of several CG pairs within the *IL17A* promoter region were elevated in SLE CD4$^+$ T cells transfected with an RFX1 expression plasmid compared with empty control (Fig. 4d–f).

Our previous studies showed that RFX1 could recruit a repressor complex including DNMT1, HDAC1, and SUV39H1 proteins and that RFX1 downregulation could lead to decreased recruitment of DNMT1, HDAC1, and SUV39H1 to the promoters of *CD11a* and *CD70* in SLE CD4$^+$ T cells[16, 21]. Therefore, we determined whether RFX1 also modulates the enrichments of DNMT1, HDAC1, and SUV39H1 in the *IL17A* promoter in CD4$^+$ T cells. ChIP-qPCR assays showed that the binding levels of all three proteins were decreased significantly when RFX1 was knocked down in normal CD4$^+$ T cells (Fig. 4g–i), whereas the binding of all three proteins to the *IL17A* promoter was increased significantly when RFX1 was overexpressed in SLE CD4$^+$ T cells (Fig. 4j–l).

**RFX1 regulates the differentiation of Th17 cells**. To explore the role of RFX1 during the differentiation of Th17 cells, we isolated

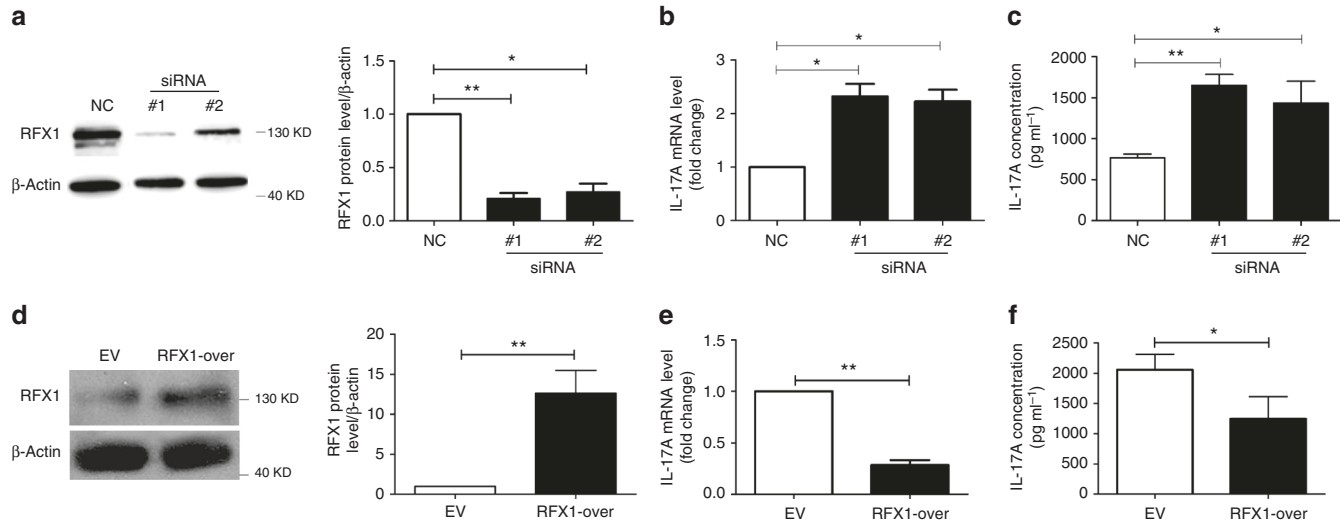

**Fig. 2** RFX1 inhibits IL-17A expression in CD4[+] T cells. **a** CD4[+] T cells from healthy controls were transfected with negative control siRNA (NC), RFX1-specific siRNA#1 or siRNA#2, along with anti-CD3 and -CD28 antibody stimulation. The RFX1 protein level was downregulated significantly in CD4[+] T cells transfected with siRNA#1 or siRNA#2 compared with NC. Representative results of western blot analysis are shown in the left panel, and the quantification of three independent experiments is shown in the right panel. **b** mRNA expression levels of IL-17A in cells from three groups were quantified by RT-qPCR. **c** IL-17 protein levels in the supernatants of culture media of three groups were measured by ELISA. **d** CD4[+] T cells from SLE patients were transfected with RFX1 expression plasmid or empty control plasmid. Representative results of western blot analysis are shown in the left panel and the quantification of three independent experiments is shown in the right panel. **e** mRNA expression levels of IL-17A in cells of two groups were quantified by RT-qPCR. **f** IL-17A protein levels in supernatants of culture media of two groups were measured by ELISA. Data are representative of three independent experiments (mean ± s.d.; n = 3). *P < 0.05 and **P < 0.01, compared between the indicated groups. P-values were determined using the two-tailed Student's t-test

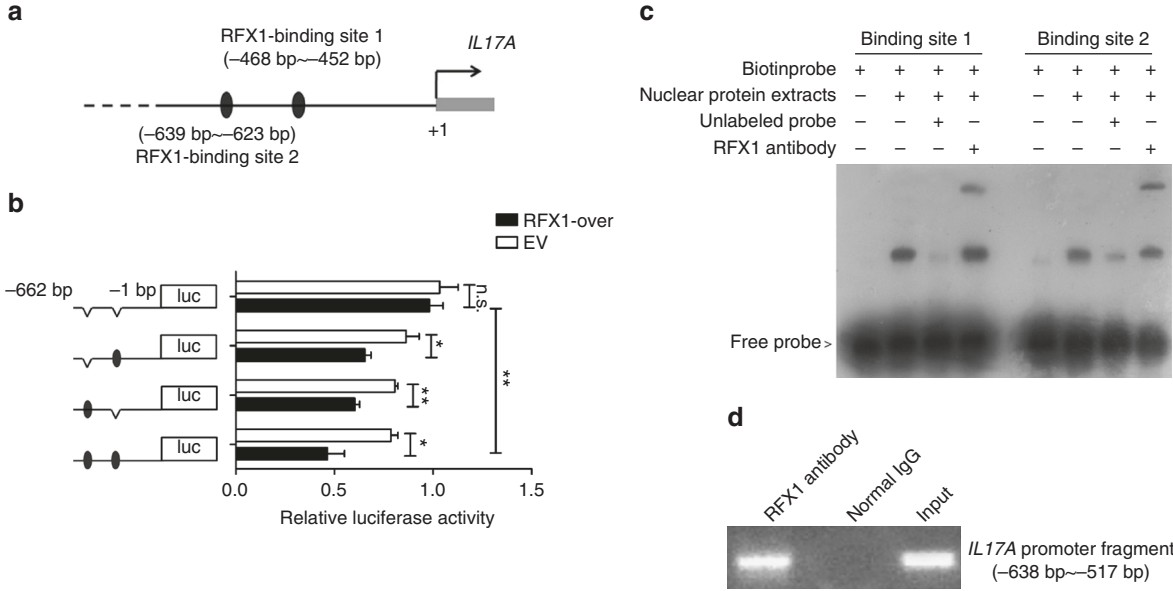

**Fig. 3** RFX1 regulates IL-17A expression through binding to the IL17A promoter. **a** The two predicted binding sites of RFX1 in the IL17A promoter. Site 1 is located at ~468–452 bp, and Site 2 is located at ~639–623 bp, both upstream of the TSS of IL17A. **b** Luciferase reporter vectors with or without binding Site 1 and/or Site 2 are shown to the left. Jurkat cells were transfected with the reporter plasmids and either pcDNA3 empty vector (white bars) or RFX1 expression plasmid (black bars). Cells were lysed 48 h after transfection, and firefly luciferase activity was measured and normalized to Renilla luciferase activity. Each experiment was performed in Jurkat cells in triplicate (n = 3), and values are given as the mean ± s.d. *P < 0.05; **P < 0.01; n.s., not significant, compared between the indicated groups. P-values were determined using two-tailed Student's t-tests. **c** EMSA was used to detect the binding of RFX1 protein with binding Site 1 and Site 2 in the IL17A promoter in vitro. The biotin probe specific for Site 1 or Site 2 was bound by nuclear protein extracts of CD4[+] T cells, which could be blocked by an RFX1 antibody. **d** ChIP assay was used to analyze the in vivo relevance of RFX1 binding to the IL17A promoter. Gel electrophoresis bands indicated the binding of RFX1 in the promoter region of IL17A. Three independent experiments were repeated in **c**, **d**

naive CD4[+] T cells from human peripheral blood and induced Th17 cell differentiation under Th17-polarizing conditions. We found that the mRNA and protein levels of RFX1 were significantly downregulated in induced Th17 cells on days 3 and 5

compared with naive CD4[+] T cells (Supplementary Fig. 2a, b). Furthermore, we transfected both an RFX1 expression plasmid and a blank control into naive CD4[+] T cells and then induced Th17 differentiation under Th17-polarizing conditions. The

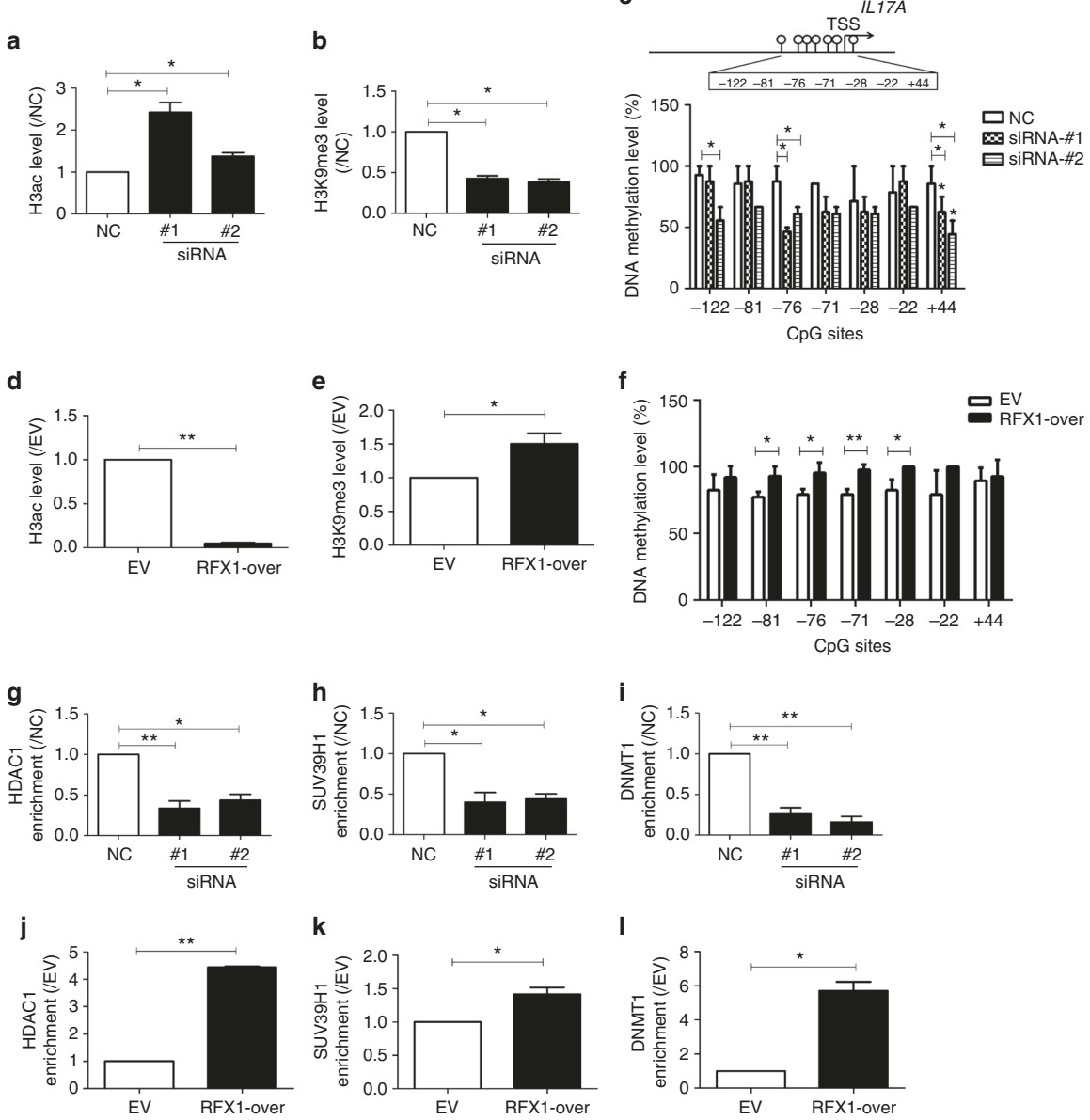

**Fig. 4** RFX1 regulates histone modifications and DNA methylation of the *IL17A* gene. **a**, **b** Changes in histone H3 acetylation (**a**) and H3K9me3 (**b**) in the promoter of *IL17A* in normal CD4+ T cells transfected with siRNAs (siRNA#1 and siRNA#2) relative to NC were analyzed by ChIP-qPCR. **c** DNA methylation level of the human *IL17A* proximal promoter was measured by BSP. **d**, **e** Fold-changes of histone H3 acetylation (**d**) and H3K9me3 (**e**) in the promoter of *IL17A* in SLE CD4+ T cells transfected with RFX1 expression plasmid (RFX1-over) or empty vector (EV). **f** DNA methylation level of the human *IL17A* proximal promoter was measured by BSP. Panels **g–l** demonstrate the changes in HDAC1 (**g**, **j**), SUV39H1 (**h**, **k**), and DNMT1 (**i**, **l**) enrichment in the promoter of *IL17A* in normal CD4+ T cells transfected with siRNAs (siRNA#1 and siRNA#2) relative to NC or in SLE CD4+ T cells transfected with RFX1 expression plasmid (RFX1-over) relative to empty vector (EV). The levels in NC or EV control groups were set to "1". The fold-changes were calculated by the ratio of RFX1-over or siRNAs to EV or NC. Data are representative of three independent experiments (mean ± s.d.; n = 3). *P < 0.05 and **P < 0.01, compared with the indicated groups. P-values were determined using two-tailed Student's t-tests. *IL17A*-Primer1 shown in Supplementary Table 4 was used for ChIP-qPCR

results of flow cytometric analysis indicated that the percentage of IL-17+CD4+ T cells was decreased in cells transfected with the RFX1 expression plasmid compared with the blank control (Supplementary Fig. 2c). Previous studies have shown that the *IL17A* promoter is demethylated and the H3K27 tri-methylation level in the *IL17A* promoter is decreased in Th17 cells[24, 25]. Here we analyzed the enrichment of RFX1 and the levels of H3ac and H3K9me3 in the *IL17A* promoter in induced Th17 cells. ChIP-qPCR results showed that the binding of RFX1 was significantly decreased, along with increased H3ac level and decreased

H3K9me3 level in the induced Th17 cells on day 5 compared with naive CD4+ T cells (Supplementary Fig. 3a–c). However, no significant change was observed in H3K4m3 level (Supplementary Fig. 3d).

To determine whether RFX1 downregulation in pathogenic CD4+ T cells is a functionally relevant regulator of the development of autoimmune inflammation, we used homologous recombination to generate mice with an *Rfx1* allele flanked by loxP sites (floxed; Fig. 5a). Mice with confirmed germline transmission were crossed with CD4-cre transgenic mice to

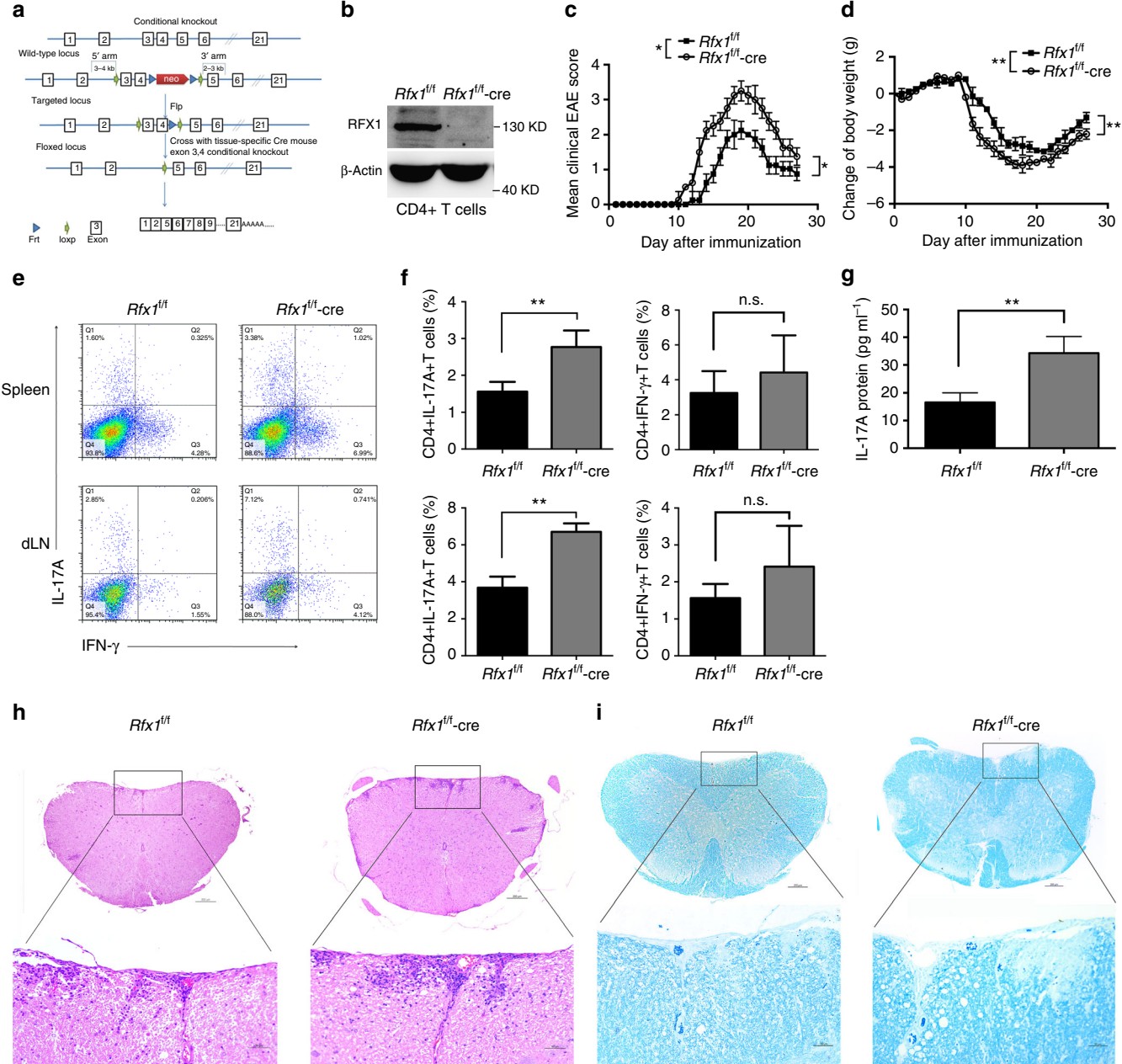

**Fig. 5** EAE is aggravated in *Rfx1* cKO mice. **a** Schematic representation of the *Rfx1* locus and targeting strategy. Cre-mediated recombination of loxP sites in mice. Exon 3 and 4 of *Rfx1* were knocked out in *Rfx1*^f/f^ by CD4-cre, which was named after *Rfx1*^f/f^-cre (cKO). **b** Representative western blot analysis of RFX1 protein expression in CD4^+ T cells of *Rfx1*^f/f^-cre and control *Rfx1*^f/f^ (*n* = 3 mice per group). **c, d** Clinical scores and weight loss of *Rfx1*^f/f^ or *Rfx1*^f/f^-cre mice after the induction of EAE were assessed every day (*n* = 5 mice per group). **e, f** Flow cytometric analysis of the percentages of CD4^+IL-17A^+ T cells and CD4^+IFN-γ^+ T cells in the spleens and draining lymph nodes (dLN) derived from *Rfx1*^f/f^ or *Rfx1*^f/f^-cre mice (*n* = 5 mice per group). **g** The concentration of IL-17A protein in serum from *Rfx1*^f/f^ and *Rfx1*^f/f^-cre mice with EAE was measured by ELISA (*n* = 5 mice per group). **h** Infiltrating lymphocytes in the spinal cord of *Rfx1*^f/f^ and *Rfx1*^f/f^-cre mice with EAE were analyzed by H&E staining. **i** The degree of demyelination of spinal cords from *Rfx1*^f/f^ and *Rfx1*^f/f^-cre mice with EAE was analyzed by luxol fast blue staining (*n* = 5 mice per group). Scale bars, 200 or 50 μm (magnified panels). Data are shown as mean ± s.d. *P < 0.05; **P < 0.01; n.s., not significant, two-way analysis of variance and Mann–Whitney *U*-test (two-tailed) for **c, d**, and two-tailed Student's *t*-tests for **f, g**

generate a conditional knockout mouse model with Rfx1 expression deficit specifically in CD4^+ T cells (Fig. 5b and Supplementary Fig. 4). These mice remained healthy without any detectable immune-mediated pathology for at least 32 weeks. EAE is a typical Th17-related autoimmune disease modeled in mice[26–28], and Rfx1 expression was decreased significantly in CD4^+ T cells from EAE mice compared with wild type (WT) mice (Supplementary Fig. 5). Therefore, EAE disease models in Rfx1-sufficient (*Rfx1*^f/f^) and Rfx1-deficient (*Rfx1*^f/f^-cre) mice were

induced by immunization with myelin oligodendrocyte glyco-protein (MOG35-55). *Rfx1*^f/f^ mice developed signs of EAE on day 12 and reached the peak of the disease on day 19 after immunization. By contrast, disease onset advanced and was significantly more severe in *Rfx1*^f/f^-cre mice when quantified by clinical score or weight loss (Fig. 5c, d). To determine whether Rfx1 inhibition aggravated EAE by increasing the differentiation of Th17 cells, we immunized *Rfx1*^f/f^ and *Rfx1*^f/f^-cre mice with MOG35-55 and quantified the frequencies of IL-17A- and

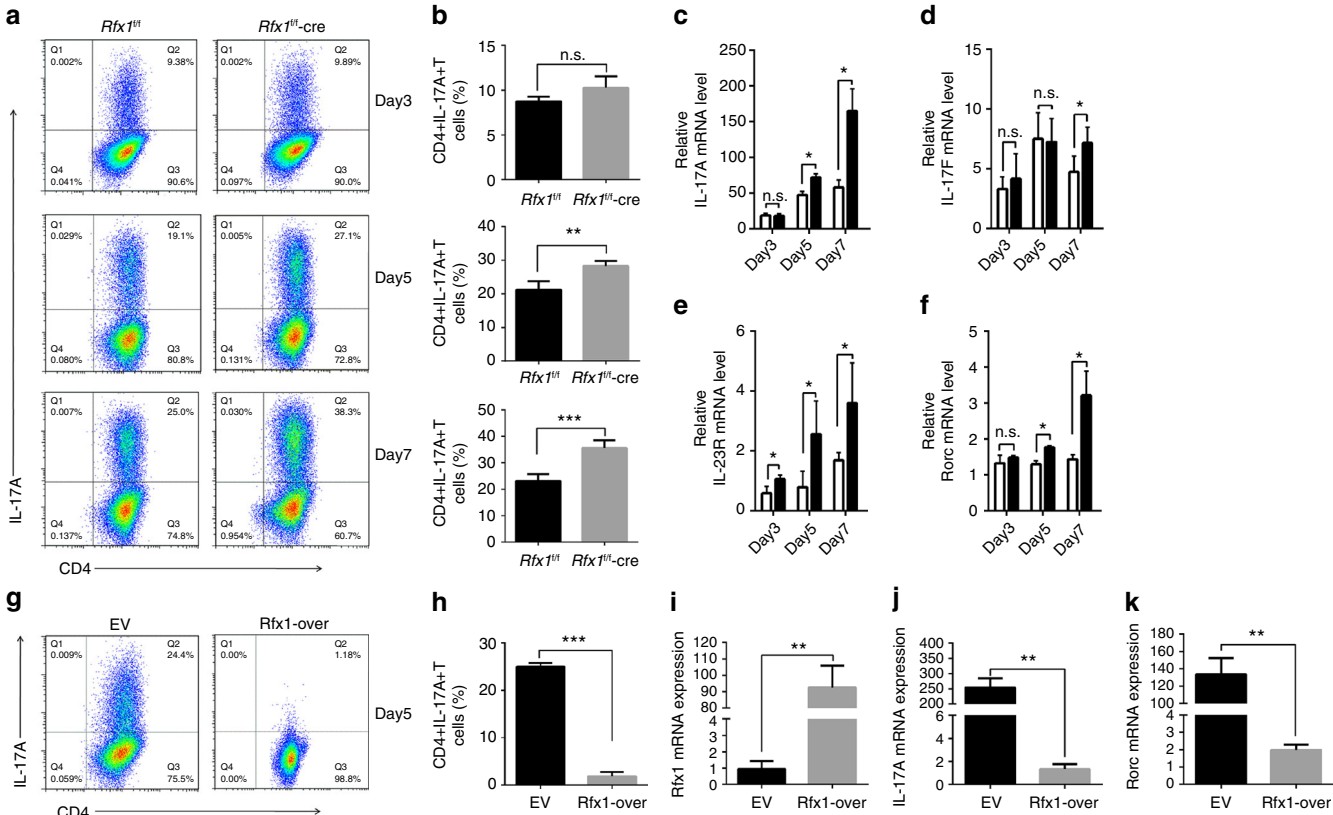

**Fig. 6** RFX1 restrains Th17 cell differentiation in vitro. **a** Representative flow cytometric analysis of Th17 cells polarized from naive T cells of $Rfx1^{f/f}$ and $Rfx1^{f/f}$-cre mice on days 3, 5, and 7 in the left panel. **b** shows the quantification (mean ± s.d.; $n = 4$ mice per group). **c–f** mRNA expression levels of Th17-related genes, including IL-17A (**c**), IL-17F (**d**), IL-23R (**e**), and Rorc (**f**) (mean ± s.d.; $n = 4$). **g**, **h** Flow cytometric analysis of in vitro-induced Th17 cells from naive T cells of $Rfx1^{f/f}$-cre mice transfected with mouse Rfx1 expression vector (Rfx1-over) or empty expression vector (EV) 5 days after transfection. **i–k** mRNA expression levels of Rfx1 (**i**), IL-17A (**j**), and Rorc (**k**) genes in Th17 cells induced in vitro. Data are representative of three independent experiments (mean ± s.d.; $n = 3$). $*P < 0.05$; $**P < 0.01$; $***P < 0.001$; n.s., not significant, two-tailed Student's $t$-test

interferon (IFN)-γ-producing cells and the production of IL-17A in serum 19 days after immunization. As predicted by our in vitro results, Rfx1 deficiency significantly increased the percentage of CD4+IL-17A+ T cells in the spleens and draining lymph nodes of immunized mice, as well as the production of IL-17A in serum, without affecting the production of IFN-γ (Fig. 5e–g). Independent histological analysis of spinal cords demonstrated significantly increased inflammation and demyelination in $Rfx1^{f/f}$-cre mice (Fig. 5h, i). In addition, we isolated splenocytes from the spleen of $Rfx1^{f/f}$ and $Rfx1^{f/f}$-cre mice immunized with MOG35-55 at day 19 and further cultured with MOG (30 μg) re-stimulation in vitro. After 3 days, IL-17A and IFN-γ concentrations in the supernatant were measured by enzyme-linked immunosorbent assays (ELISA). The results showed that IL-17A production was increased significantly in $Rfx1^{f/f}$-cre mice compared with $Rfx1^{f/f}$ mice, whereas IFN-γ production was not significantly different between two groups (Supplementary Fig. 6). This result further points to the importance of Rfx1 in the generation of Th17 cells and the production of IL-17.

To confirm that Rfx1 plays a role in Th17 cell differentiation, we isolated naive CD4+ T cells from $Rfx1^{f/f}$ or $Rfx1^{f/f}$-cre mice and stimulated them under Th17-polarizing conditions. As shown in Fig. 6a, b, the absence of Rfx1 resulted in a significant increase in the percentage of IL-17-producing T cells on days 5 and 7. The significantly elevated mRNA expression levels of IL-17A, IL-17F, IL-23R, and Rorc also indicated increased Th17 cell differentiation in Rfx1-deficient mice compared with Rfx1-sufficient mice (Fig. 6c–f). To expand these results, we transfected

naive CD4+ T cells from $Rfx1^{f/f}$-cre mice with either an empty vector or an Rfx1 expression plasmid. This allowed us to evaluate Rfx1-deficient cells before and after Rfx1 reconstitution. Cells were then stimulated in Th17-polarizing conditions, and the frequency of IL-17A-producing cells was quantified by flow cytometry (Fig. 6g). As expected, the absence of Rfx1 resulted in a significant increase in CD4+IL-17A+ T cells. However, the reconstitution of Rfx1 restored the inhibition of Th17 differentiation significantly (Fig. 6h–k).

Next we verified the regulation by Rfx1 of the chromatin modifications of $Il17a$ in Rfx1-deficient mice and EAE mice. We isolated CD4+ T cells from the spleens of $Rfx1^{f/f}$ and $Rfx1^{f/f}$-cre mice. After 24 h of activation with anti-CD3 and anti-CD28 antibodies, ChIP-qPCR and bisulfite sequencing PCR (BSP) was used to determine the binding of HDAC1, SUV39H1, and DNMT1 and the levels of H3ac, H3K9me3, and DNA methylation in the promoter region of the mouse $Il17a$ (m$Il17a$) gene containing the RFX1-binding motif. As expected, the binding of HDAC1, SUV39H1, and DNMT1 in the m$Il17a$ promoter was significantly decreased in $Rfx1^{f/f}$-cre mice compared with $Rfx1^{f/f}$ mice, leading to an increased H3ac level and decreased H3K9me3 and DNA methylation levels in the m$Il17a$ promoter (Supplementary Fig. 7). Similar changes in chromatin modifications were also observed in CD4+ T cells from EAE mice compared with WT mice (Supplementary Fig. 8) Therefore, we concluded that Rfx1 deficiency in CD4+ T cells led to the increased Th17 response through the induction of an open chromatin status in the $Il17a$ gene.

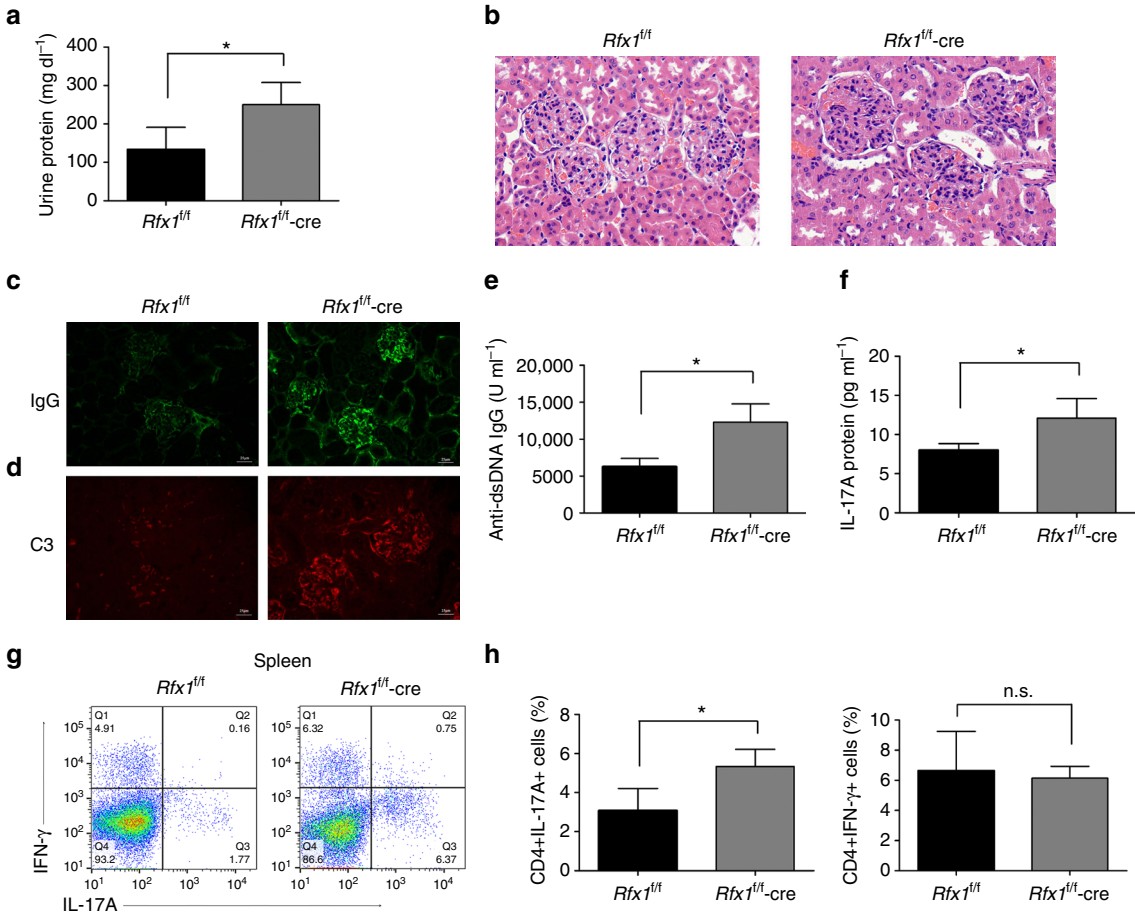

**Fig. 7** *Rfx1* cKO mice have more severe lupus autoimmunity. *Rfx1*^f/f-cre (*n* = 4) or *Rfx1*^f/f (*n* = 3) mice were injected with 500 μl of pristane, respectively, as described in Materials and Methods section. Mice were sacrificed at 20 weeks after pristane treatment. **a** Urine protein excretion was measured at the end of the observed period. **b** Representative images of hematoxylin and eosin-stained sections of kidneys collected at the end of observed period are shown (magnification ×400). IgG (**c**) and C3 (**d**) depositions in the kidney sections were assessed by immunofluorescence. Scale bar, 25 μm. Serum anti-dsDNA IgG (**e**) and IL-17A protein (**f**) were quantified by ELISA. **g**, **h** Flow cytometric analysis of the percentages of CD4⁺IL-17A⁺ T cells and CD4⁺IFN-γ⁺ T cells in the spleens of *Rfx1*^f/f-cre mice and *Rfx1*^f/f mice were evaluated by flow cytometry. Data are shown as mean ± s.d. *P < 0.05; n.s., not significant, two-tailed Student's *t*-tests

**Rfx1 deficiency exacerbates lupus disease.** To further test whether Rfx1 deficiency exacerbates lupus disease, *Rfx1*^f/f-cre mice and *Rfx1*^f/f control mice were injected with pristane to induce lupus-like model and observed for 5 months. At the end of the observation period, urinalysis showed that *Rfx1*^f/f-cre mice had a higher proteinuria level compared to *Rfx1*^f/f control mice (Fig. 7a). Hematoxylin and eosin (H&E) staining of kidney sections revealed that *Rfx1*^f/f-cre mice had a marked increase in pathological lesions of glomerulus compared to *Rfx1*^f/f control mice (Fig. 7b). Immunostaining analysis revealed more deposition of immunoglobulin G (IgG) and complement 3 (C3) in the glomeruli of *Rfx1*^f/f-cre mice compared with that in *Rfx1*^f/f control mice (Fig. 7c, d). Serum levels of anti-dsDNA antibody and IL-17A protein were also significantly increased in *Rfx1*^f/f-cre mice (Fig. 7e, f). In addition, flow cytometry showed an increased percentage of CD4⁺IL-17A⁺ T cells in the spleen of *Rfx1*^f/f-cre mice compared to *Rfx1*^f/f control mice (Fig. 7g, h). Collectively, Rfx1 deficiency promotes the development of lupus-like mice model and exacerbates renal damage.

**RFX1 is a downstream of STAT3.** To clarify if the Th17-polarizing cytokines repress the expression of RFX1, we measured RFX1 expression in CD4⁺ T cells with transforming growth factor (TGF)-β, IL-6, IL-23, or IL-1β stimulation respectively. The

mRNA and protein expression levels of RFX1 were significantly inhibited by IL-6 stimulation, but no significant change was induced by TGF-β, IL-23, or IL-1β alone (Fig. 8a, b). Decreased RFX1 level was significantly restored by STAT3 inhibitor, indicating that IL-6-STAT3 signal is necessary for RFX1 repression under Th17-polarizing conditions (Fig. 8c, d). In addition, we also found that RFX1 expression level was negatively correlated with serum C-reactive protein (CRP) level, a surrogate marker of IL-6 activity, in SLE patients with active arthritis (Supplementary Fig. 9).

To investigate the mechanism of RFX1 downregulation in CD4⁺ T cells of SLE patients, we also determined the effect of IL-6-STAT3 signaling on RFX1 expression in CD4⁺ T cells of SLE patients. We compared the phosphorylated STAT3 (pSTAT3) level between SLE patients and healthy controls. As expected, the pSTAT3 level was higher in CD4⁺ T cells of active and inactive SLE patients than in healthy controls (Supplementary Fig. 10a). RFX1 expression was upregulated in CD4⁺ T cells of SLE patients when STAT3 expression was knocked down with siRNA (Fig. 8e and Supplementary Fig. 10b, c). In addition, we found that RFX1 expression was repressed in normal CD4⁺ T cells transfected with a STAT3 expression plasmid under IL-6 stimulation (Supplementary Fig. 10d–f).

Enhancer activity of the seventh intron of the *RFX1* gene has been reported[29]. The isolated CpG island located in the seventh

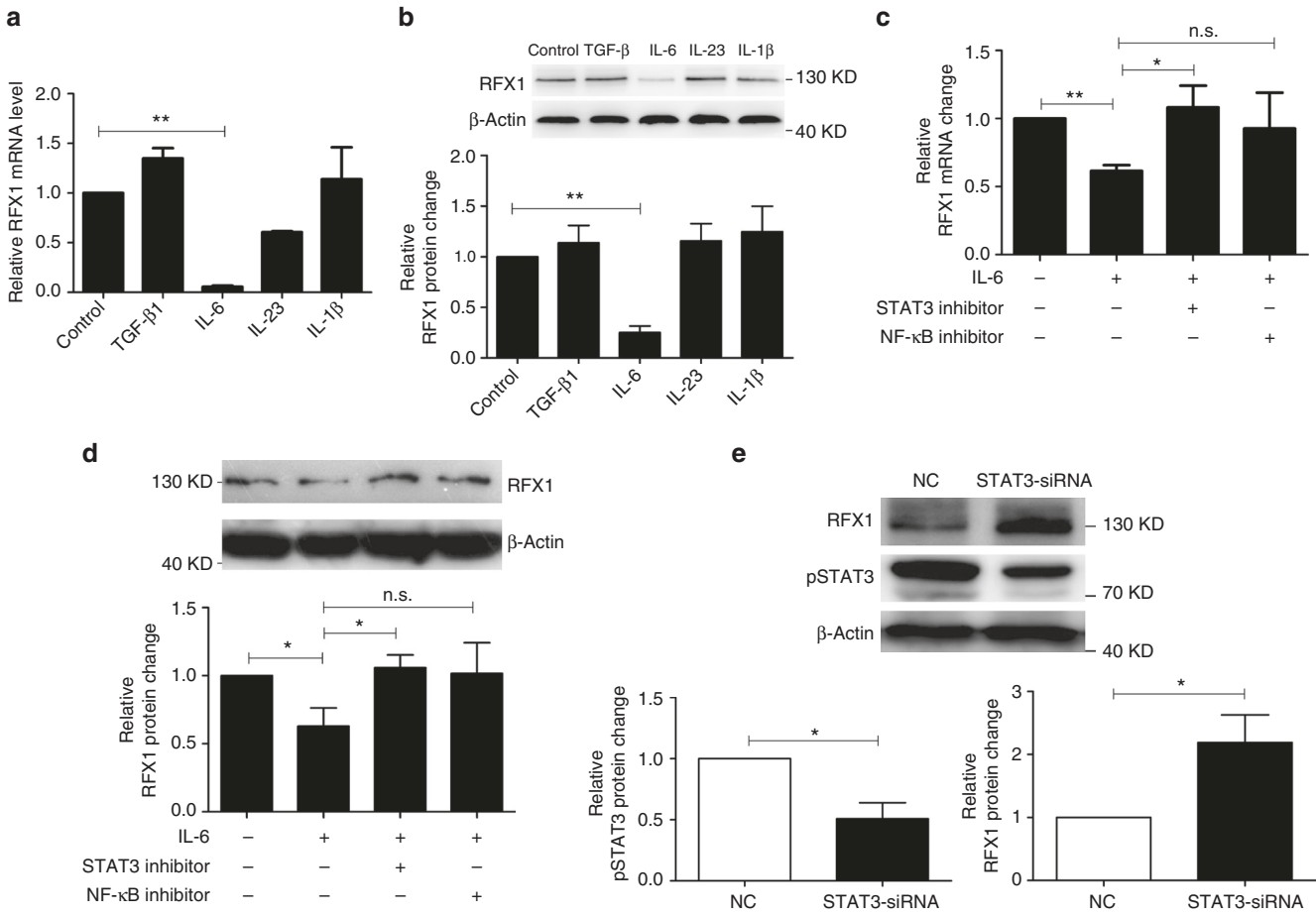

**Fig. 8** The IL-6-STAT3 pathway inhibits RFX1 expression. **a**, **b** qPCR and western blot analysis of RFX1 mRNA and protein expression levels in CD4+ T cells upon TGF-β, IL-6, IL-23, or IL-1β stimulation. **c**, **d** qPCR and western blot analysis of RFX1 mRNA and protein expression levels in CD4+ T cells stimulated by IL-6 along with STAT3 inhibitor or NF-κB inhibitor. The y axis indicates the fold-change in RFX1 expression compared with a negative control. **e** Western blot analysis of RFX1 and phosphorylated STAT3 (pSTAT3) in CD4+ T cells transfected with STAT3 siRNA or negative control (NC) at 48 h after transfection. Negative control was set as "1". The y axis indicates the fold-change in RFX1 expression compared with a negative control. The protein expression level is calculated by the ratio of protein to β-actin. Data are representative of three independent experiments (mean ± s.d.; $n = 3$). *$P < 0.05$, **$P < 0.01$, compared between the indicated groups. n.s., not significant. P-values were determined using two-tailed Student's t-tests

intron of the *RFX1* gene was hypermethylated in glioma cell lines and tissues but not in normal brain tissue or lymphocytes[29]. We predicted three potential STAT3-binding sites in the seventh intron of the *RFX1* gene (Fig. 9a). ChIP-qPCR analysis showed that the enrichment of pSTAT3 was increased in the seventh intron of the *RFX1* gene in CD4+ T cells under IL-6 stimulation, but no significant change was observed without IL-6 stimulation (Fig. 9b). Moreover, we constructed a luciferase reporter plasmid containing a DNA fragment with STAT3-binding sites (WT) or with mutant STAT3-binding sites, both separately and together (Mu). STAT3 overexpression repressed the luciferase activity in the WT reporter construct or Mu reporter construct with a mutant signal site. However, the repression could be removed when all three sites were mutated simultaneously (Fig. 9c). In addition, we found that the luciferase activity was lower in CD4+ T cells under IL-6 stimulation without or with STAT3 over-expression compared with negative control or empty vector control (Fig. 9d, e). These data indicated that IL-6-induced STAT3 phosphorylation represses RFX1 expression through reducing the enhancer activity of the seventh intron of the *RFX1* gene in CD4+ T cells.

A previous study indicated that STAT3 can repress gene transcription by regulating chromatin modification through the recruitment of HDAC1 and DNMT1[30, 31]. To determine whether

IL-6-STAT3 signaling has an effect on the chromatin conformation in the regulatory region of the *RFX1* gene, we measured the DNA methylation status and histone modification of the seventh intron of the *RFX1* gene. ChIP-qPCR results showed that IL-6 stimulation significantly reduced the H3 and H4 acetylation levels in the seventh intron of the *RFX1* gene in CD4+ T cells (Fig. 9f). As the methylation of CpG island located in the seventh intron of the *RFX1* gene has an effect on the enhancer activity of the seventh intron, we measured the methylation level of CpG island in the seventh intron of the *RFX1* gene and found that IL-6 stimulation elevated the methylation level in CD4+ T cells (Fig. 9g). In addition, decreased H3 acetylation and DNA hypermethylation in the seventh intron of the *RFX1* gene were also observed in SLE CD4+ T cells compared with healthy controls (Supplementary Fig. 11). These data suggest that RFX1 downregulation is due to heterochromatin modifications induced by IL-6 stimulation in CD4+ T cells of SLE patients.

## Discussion

Transcription factor-mediated epigenetic modifications regulate the spatiotemporal expression of genes in the development and differentiation of cells, as well as in the occurrence and progression of diseases[32]. Our previous study was the first to find a role

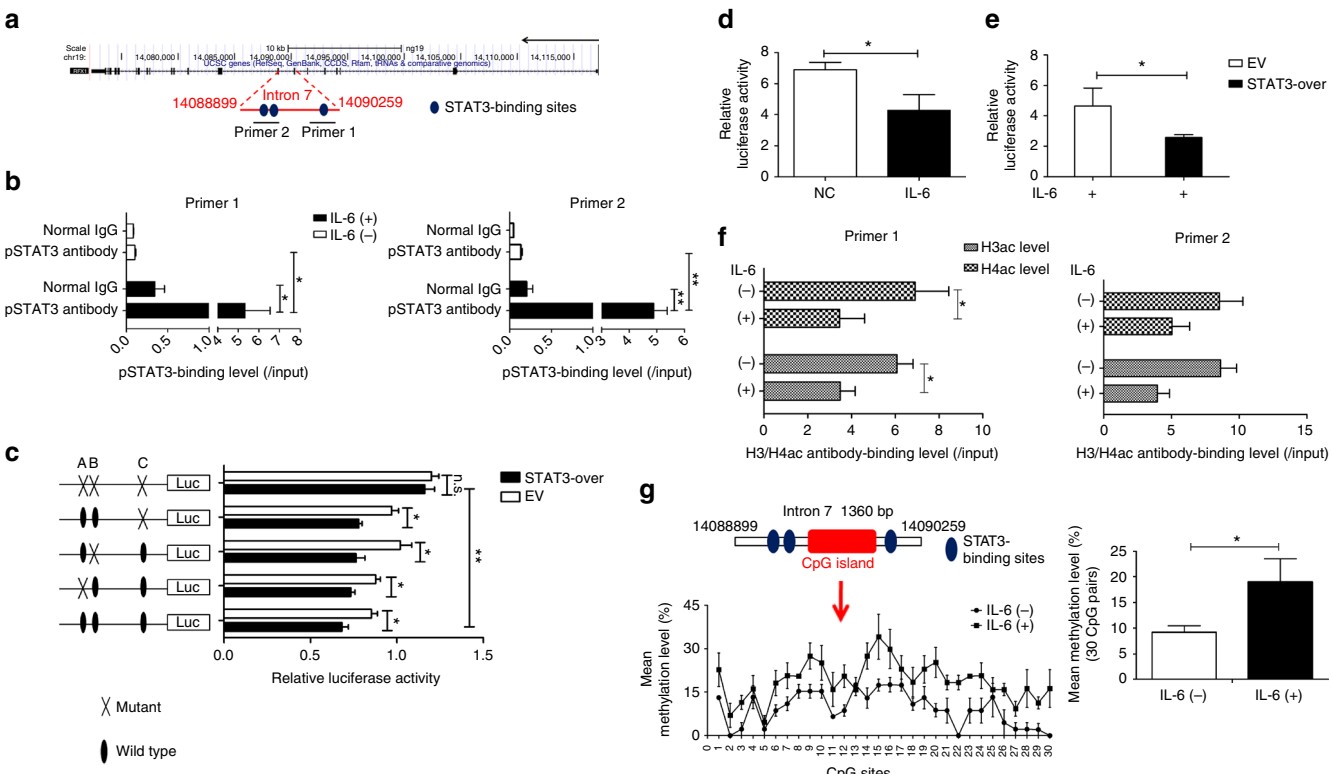

**Fig. 9** The IL-6-STAT3 pathway regulates epigenetic modifications of the *RFX1* locus. **a** Genome location of the *RFX1* locus and the STAT3-binding sites in intron 7 of the *RFX1* gene. **b** ChIP-qPCR assay of pSTAT3 binding in intron 7 of *RFX1* in CD4[+] T cells with or without IL-6 stimulation. Primer 1 and Primer 2 were used to amplify the fragment containing STAT3-binding sites. **c** Luciferase reporter assay analysis of STAT3 regulation on the enhancer activity of intron 7 in 293T cells transfected with STAT3 expression vector (STAT3-over) or empty expression vector (EV). A, B, and C represent the STAT3-binding sites in intron 7. **d** Luciferase reporter assay analysis of the enhancer activity of intron 7 in CD4[+] T cells with IL-6 stimulation and negative control. **e** Luciferase reporter assay analysis of the enhancer activity of intron 7 in CD4[+] T cells with IL-6 stimulation transfected with STAT3 expression vector (STAT3-over) or empty expression vector (EV). **f** ChIP-qPCR analysis of H3 acetylation level and H4 acetylation level of intron 7 of the *RFX1* gene in CD4[+] T cells with or without IL-6 stimulation. **g** BSP analysis of the DNA methylation level of CpG island in intron 7 of the *RFX1* gene in CD4[+] T cells with or without IL-6 stimulation. The left panel shows the methylation level of each CpG site, and the right panel shows the mean methylation level of all 30 CpG sites in the CpG island. Data are representative of three independent experiments (mean ± s.d.; *n* = 3). *$P < 0.05$, **$P < 0.01$, compared between the indicated groups. n.s., not significant. *P*-values were determined using two-tailed Student's *t*-tests

for transcription factor RFX1 in the pathogenesis of SLE. RFX1 binds to the promoter regions of the *CD11a* and *CD70* genes and forms a protein complex via the recruitment of HDAC1, DNMT1, and SUV39H1 to regulate the histone acetylation and methylation and DNA methylation of the *CD11a* and *CD70* promoters in CD4[+] T cells. The downregulation of RFX1 leads to the overexpression of CD11a and CD70 in SLE CD4[+] T cells, which induces the increased reactivity of T cells[16, 21]. In the present study, we found that RFX1 expression was significantly negatively correlated with the expression of IL17A, as well as CD11a and CD70 expression, in CD4[+] T cells of patients with SLE.

Recently, a growing body of evidence has supported the role of the effector cytokine IL-17A and Th17 cells in the pathogenesis of SLE[33]. It has been noted that patients with SLE, including those with new-onset disease, display increased serum or plasma levels of IL-17A, have an expansion of IL-17-producing T cells in the peripheral blood, and an infiltration of Th17 cells in the target organs, including the kidneys[34, 35]. Additionally, increased production of IL-17A correlates with disease activity in patients with SLE[5]. Here we demonstrated an important novel role for RFX1 in the regulation of the increased IL-17A expression in CD4[+] T cells of patients with SLE, as well as in the differentiation of Th17 cells and the development of the IL-17-related autoimmune diseases EAE and lupus in mice. We provide evidence demonstrating that

downregulation of RFX1 contributes to IL-17A overexpression and promotes IL-17A-producing T cell differentiation in SLE patients.

Here we present evidence that RFX1 is a negative regulatory factor in Th17 differentiation and autoimmune diseases. The expression of Th17-related genes such as IL-17A, IL-17F, IL-23R, and Rorc was increased in induced Th17 cells with Rfx1 deficiency, suggesting that differentiation of Th17 cells was promoted by the loss of Rfx1 under Th17-polarizing conditions. Moreover, lack of Rfx1 increased EAE clinical and histopathologic severity scores, as well as the frequency of peripheral IL-17A-producing cells. T cells from the spleens and draining lymph nodes of Rfx1-deficient mice displayed no significant changes in IFN-γ production, but did have an overproduction of IL-17A, highlighting a predominant inhibitory effect of Rfx1 on Th17 differentiation. In addition, we also induced lupus-like mice model by intraperitoneal injection of pristane in Rfx1-deficient mice. Rfx1 deficiency increased the serum levels of anti-dsDNA antibody and IL-17A protein. Importantly, the more severe lupus-like kidney damage and deposition of IgG and C3 in kidney were observed in mice with Rfx1 deficiency. These results underscore the importance of RFX1 in the differentiation of Th17 cells and the pathogenesis of autoimmune diseases. Of interest, we found that IL-17F and RORC expression had a somewhat small reduction, but no significant difference, in CD4[+] T cells of SLE patients

compared with normal controls, which is discrepant with the increased IL-17F and Rorc expression in the induced Th17 cells from mice with Rfx1 deficiency. The results may be due to more complicated regulatory mechanism of IL-17F expression in CD4⁺ T cells of SLE patients, which is different from the mechanism under Th17-polarizing conditions.

DNA methylation and histone modifications, such as acetylation and methylation, play an important role in regulating gene expression through affecting the genomic accessibility of transcription factors[36]. It has been reported that the *IL17A* gene in human T cells from SLE patients and Th17 cells is subject to epigenetic remodeling at various levels, including DNA hypomethylation and reduced H3K27 tri-methylation in the promoter region of the *IL17A* gene[15, 37, 38]. In this study, we identified increased H3 acetylation (euchromatin) and decreased H3K9 tri-methylation (heterochromatin) in the promoter of the *IL17A* gene in SLE CD4⁺ T cells compared with healthy controls and in induced Th17 cells compared with naive CD4⁺ T cells. In addition, we demonstrated that the downregulation of transcription factor RFX1 caused the DNA hypomethylation, the increased H3 acetylation, and the decreased H3K9 tri-methylation through the reduced recruitment of HDAC1, SUV39H1, and DNMT1 to the promoter region of *IL17A* in SLE CD4⁺ T cells. The epigenetic regulation of the *Il17a* gene was also confirmed in CD4⁺ T cells from mice deficient in Rfx1. The level of H3 acetylation was increased, while the levels of H3K9 tri-methylation and DNA methylation were decreased in the *Il17a* locus; meanwhile, the recruitment of HDAC1, SUV39H1, and DNMT1 was reduced significantly. Although the global DNA hypomethylation and aberrant histone modifications in the chromatin of CD4⁺ T cells have been identified in SLE patients[39], the site-specific epigenetic modifications mediated by transcription factor RFX1 in *IL17A*, *CD11a*, and *CD70* loci[16, 21] facilitated the characterization of altered epigenetic regulation occurring in the chromatin of CD4⁺ T cells, leading to the activation and inactivation of gene transcription. In fact, transcription factor-mediated aberrant epigenetic modifications in SLE have been reported in a previous study[15, 40]. For example, CREM interacts with DNMT3a, which mediates de novo CpG-DNA methylation, and facilitates DNMT3a recruitment to the CRE site of the human *IL2* promoter, thereby repressing the transcription of the *IL2* gene[40].

Th17 differentiation requires the presence of IL-6 and its signal transduction partner STAT3[41]. Our findings provide evidence that RFX1 is a downstream partner of the IL-6/STAT3 signaling pathway. Moreover, reduction of RFX1 during Th17 differentiation is independent of TGF-β, IL-1β, and IL-23, as RFX1 expression was downregulated significantly in CD4⁺ T cells with IL-6 stimulation alone, which could be restored in the presence of STAT3 inhibitors. STAT3, activated by IL-6 and IL-23, is an essential regulator of lineage commitment of Th17 cells through the activation of *RORC* transcription by binding to the regulatory region[42]. By contrast, STAT3 can also repress gene transcription in different cell environments. Yeh et al. revealed that *NR4A3* promoter binding by STAT3 might repress its transcription by promoter DNA methylation in gastric cancer[43]. Yang et al. found that STAT3 bound to the promoter region of *Usp7* and inhibited its activity through the recruitment of HDAC1[31]. Here our data also demonstrated that the reduction of RFX1 in CD4⁺ T cells stimulated with IL-6 was due to the decreased enhancer activity of intron 7 of the *RFX1* gene induced by the increased phosphorylated STAT3 protein, which binds intron 7 of the *RFX1* gene to cause decreases in histone acetylation and increases in DNA methylation. These data revealed that the IL-6-STAT3 signaling pathway can induce Th17 differentiation through repressing RFX1 expression, suggesting a noncanonical regulatory pathway in CD4⁺ T cells, and further demonstrating the important role of

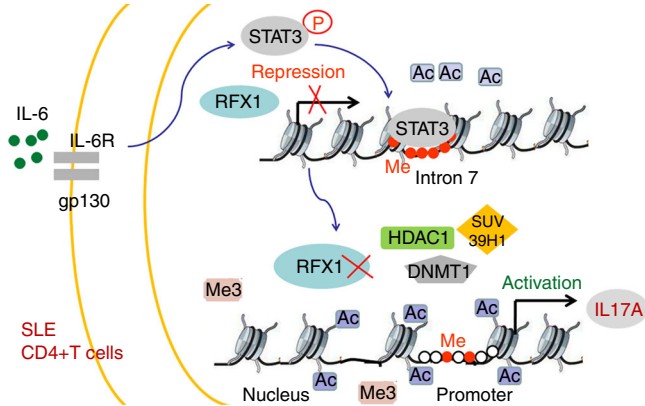

**Fig. 10** Model of an atypical IL-6-Stat3-RFX1-IL-17A regulatory pathway in CD4⁺ T cells of SLE patients. IL-6 stimulation causes the phosphorylation of STAT3, which binds intron 7 of *RFX1* to repress *RFX1* gene transcription through increased DNA methylation and decreased histone acetylation. Decreased RFX1 leads to the loss of recruitment of DNMT1, HDAC1, and SUV39H1 to the promoter of *IL17A* gene, which decreases the DNA methylation and histone H3K9 tri-methylation and increases the histone acetylation, thereby inducing the activation of *IL17A* gene transcription

transcription factors and their site-specific epigenetic regulation in the pathogenesis of SLE.

In conclusion, we have provided evidence that RFX1 downregulation contributes to the increased IL-17A expression and Th17 cell differentiation in the CD4⁺ T cells of patients with SLE, as well as promotes the pathologic changes of autoimmune diseases including EAE and lupus-like mice models. RFX1 inhibits IL-17A expression, which is largely dependent on epigenetic regulation. As depicted in Fig. 10, in normal CD4⁺ T cells, RFX1 recruits HDAC1, SUV39H1, and DNMT1 to the promoter region of *IL17A* and decreases the histone acetylation and increases the H3K9 tri-methylation and promoter hypermethylation. This results in a heterochromatin state in the promoter region of *IL17A*, leading to inactivation of *IL17A* transcription. However, in CD4⁺ T cells of patients with SLE, RFX1 expression is inhibited by IL-6-induced pSTAT3 through the regulation of the histone acetylation and DNA methylation status of intron 7 of *RFX1*, and RFX1 downregulation relieves the epigenetic restraint of the promoter region of *IL17A*, leading to the overproduction of IL-17 in SLE patients. These data demonstrate the important role of RFX1 in Th17 cells differentiation and the pathogenesis of SLE.

## Methods

**SLE patients**. This study was approved by the Ethics Committee of the Second Xiangya Hospital of Central South University, China, and written informed consent was obtained from all participants. SLE patients were recruited from outpatient dermatology clinics and in-patient wards. All patients fulfilled at least four of the SLE classification criteria of the American College of Rheumatology[44]. Lupus disease activity was assessed using the SLE Disease Activity Index (SLEDAI)[45]. Age-, sex-, and ethnicity-matched healthy individuals were enrolled as controls. Peripheral venous blood was collected from each patient and control subject and preserved with heparin. CD4⁺ T cells were isolated by positive selection using CD4 beads, according to protocols provided by the manufacturer (Miltenyi); purity was generally >95%. Basic demographic information, serum CRP levels, and SLEDAI scores at the time of blood sample collection and enrollment in our study for each patient are presented in Supplementary Table 1.

**Transfection of overexpression plasmid or siRNA**. Purified CD4⁺ T cells were cultured in 24-well plates (1 × 10⁶ cells ml⁻¹) and stimulated with plate-bound anti-CD3 antibody (5 μg ml⁻¹, Calbiochem, clone: UCHT1), followed by the addition of soluble anti-CD28 antibody (2 μg ml⁻¹, Calbiochem, clone: ANC28.1/5D10) and incubated at 37 °C for 6 h. CD4⁺ T cells were transfected with plasmid or siRNA using a Human T-cell Nucleofector Kit and an Amaxa nucleofector (Lonza). Briefly, CD4⁺ T cells were harvested, resuspended in 100 μl human T-cell nucleofector solution, and RFX1 expression plasmid pSG5-RFX1[16], STAT3

expression plasmid (Fulengen), RFX1-siRNAs (HSS109204 and HSS109206, Thermo Fisher Scientific), or STAT3-siRNA (HSS181509, Thermo Fisher Scientific) was added. Cells were electrotransfected using the nucleofector program V-024 in the Amaxa nucleofector and cultured in human T cell culture medium[16].

**RNA isolation and qPCR**. Total RNA was extracted from CD4+ T cells using Trizol reagent (Thermo Fisher Scientific). Extracted samples were reverse-transcribed with a PrimeScript® RT Reagent Kit with gDNA Eraser (TaKaRa) using 1 μg of total RNA according to the manufacturer's instructions. The reaction mixture contained 2 μl cDNA, 10 μl SYBR Premix Ex TaqTM (TaKaRa), and 400 nM sense and antisense primers in a final volume of 20 μl. Transcripts were measured using a LightCycler® 96 System (Roche). The level of gene expression was calculated using the comparative CT method. β-Actin was also amplified and used as a loading control. All primer sequences are listed in Supplementary Table 2.

**Western blotting**. CD4+ T cells were lysed and proteins were extracted and separated by sodium dodecyl sulfate-polyacrylamide gel electrophoresis using 8% polyacrylamide gels. Proteins were then transferred onto polyvinylidene difluoride membranes (Millipore). Membranes were blocked with 5% non-fat dry milk in Tris-buffered saline containing 0.1% Tween-20 (TBST) buffer and immunoblotted with antibodies against RFX1 (Santa Cruz, clone: I-19, 1:500), pSTAT3 (Tyr705, clone: D3A7, 1:2000) or β-actin (Santa Cruz, sc-130300, 1:2000). Band intensity was quantified using ImageQuantTM LAS 4000 mini (GE-Healthcare). Uncropped original scans of immunoblots were provided in Supplementary Fig. 12.

**ELISA**. The following ELISA kits were used: Human and mouse IL-17 or IFN-gamma Quantikine ELISA Kits (R&D Systems) and anti-dsDNA antibody ELISA Kit (Alpha diagnostic). All procedures were performed according to the manufacturer's instructions.

**In vitro human Th17 cells differentiation**. Peripheral blood mononuclear cells (PBMCs) were separated from the peripheral blood of healthy individuals by density gradient centrifugation (GE Healthcare). Naive T cells were isolated by negative selection using Miltenyi beads according to the manufacturer's instructions (Miltenyi Biotech). Purified naive CD4+ T cells were stimulated with plate-bound anti-CD3 (5 μg ml−1, Calbiochem, clone: UCHT1) and anti-CD28 (2 μg ml−1, Calbiochem, clone: ANC28.1/5D10) for 3 or 5 days under Th17-polarizing conditions: TGF-β (5 ng ml−1, PeproTech), IL-6 (10 ng ml−1, PeproTech), IL-1β (10 ng ml−1, PeproTech), IL-23 (20 ng ml−1, PeproTech), anti-IFN-γ (10 μg ml−1, eBioscience, clone: MD-1), and anti-IL-4 (10 μg ml−1, eBioscience, clone: MP4-25D2) for Th17. Cell culture was performed in 24-well plates in a total volume of 1 ml well−1 of culture medium with 1 × 10^6 naive CD4+ T cells. The medium was refreshed on day 3.

**In vitro mouse Th17 cells differentiation**. Naive CD4+ T cells were purified from pooled single-cell suspensions of spleens using a mouse naive CD4+ T Cell Isolation Kit II (Miltenyi) and were stimulated with plate-bound anti-CD3 (5 μg ml−1, eBioscience, clone: 145-2C11) and anti-CD28 (2 μg ml−1, eBioscience, clone: 37.51) for 3, 5, and 7 days under Th17-polarizing conditions. For Th17 cell differentiation, naive CD4+ T cells were stimulated with TGF-β (2 ng ml−1, PeproTech), IL-6 (30 ng ml−1, PeproTech), IL-1β (10 ng ml−1, PeproTech), and IL-23 (20 ng ml−1, R&D Systems) in the presence of anti-IFN-γ (10 μg ml−1, eBioscience, clone: R4-6A2) and anti-IL-4 (10 μg ml−1, eBioscience, clone: 11B11). Cell culture was performed in 24-well plates in a total volume of 1 ml well−1 of culture medium with 1 × 10^6 naive CD4+ T cells. The medium was refreshed on day 3.

**Flow cytometry**. The expression of surface markers and intracellular molecules was evaluated by flow cytometry with a FACS Canto II (BD Biosciences), and data were analyzed by the Flowjo software (Tree Star). The following antibodies were used for flow cytometry: anti-human CD4-FITC (clone: RPA-T4), IFN-γ-Percp-Cy5.5 (clone: B27), and IL-17A-PE (clone: N49-653) were purchased from BD Biosciences. Anti-mouse CD4-FITC (clone: RM4-5) was purchased from BioLegend. Anti-mouse IFN-γ-percp-Cy5.5 (clone: XMG1.2), IL-17A-PE and IL-17A-percp-Cy5.5 (clone: TC11-18H10) were purchased from BD Biosciences. To detect the expression of IFN-g and IL-17A, the cells were treated with ionomycin (750 ng ml−1, Sigma), phorbol 12-myristate 13-acetate (200 ng ml−1, Sigma), and BD GolgiPlug (BD Biosciences) for 4–6 h in a cell incubator (37 °C, 5% CO2). Surface marker staining was done at 4 °C in the dark for 30 min. All flow cytometric gating strategies are shown in Supplementary Fig. 13.

**Luciferase reporter constructs**. To generate the human IL17A promoter luciferase reporter construct, the proximal 662 bp of the human IL17A promoter (from -662 bp to -1 bp upstream of TSS) was PCR-amplified from genomic DNA of human CD4+ T cells and cloned into a luciferase vector pGL3-Basic (Promega) using the following primers: Forward, 5′-TTGGTACCTTTCAATTCCTTCCT-CAAAACACCA-3′, and Reverse, 5′-TTCTCGAGTATGAGATGGA-CAAAATGTAGCGCT-3′, with attached restriction sites for Kpn I and Xho I (underlined). The DNA fragments (from -662 bp to -1 bp upstream of TSS) with

loss of binding Site 1 and/or binding Site 2 were synthesized directly and cloned into the pGL3-Basic vector.

To generate the human RFX1 enhancer luciferase reporter construct, intron 7 containing three STAT3 binding sites of the RFX1 gene was PCR-amplified from genomic DNA of human CD4+ T cells and cloned into the luciferase vector pGL3-promoter (Promega) using the following primers: Forward, 5′-CGGGGTACCGAGGGAAATGCCTCTGTGAG-3′, and Reverse, 5′-CCGCTCGAGAGCTACAGAAAGGGCCTCAG-3′, with attached restriction sites for Kpn I and Xho I (underlined). The Mut Express II Fast Mutagenesis Kit V2 (Vazyme) was used for site-directed mutagenesis based on the template of the WT RFX1 enhancer luciferase reporter construct. The primers used to amplify the site-directed mutant plasmids are listed in Supplementary Table 3. The luciferase reporter construct with all three sites mutated was amplified based on the template of the construct with two mutant sites. All procedures were performed according to the manufacturer's instructions. All plasmid DNA preparations were carried out with DNA purification kits (Qiagen) and sequence-verified.

**Luciferase assay**. The day before transfection, 5 × 10^4 Jurkat cells (ATCC, TIB-152™) were seeded into 24-well culture plates in fresh medium. Cells were transfected with 500 ng of the IL17A promoter-luciferase reporter plasmid and 1 μg RFX1 expression plasmid or empty plasmid using the Amaxa Human T-cell Nucleofector Kit (Lonza) and an Amaxa Nucleofector II device (program U014; Lonza). HEK293T cells (ATCC, CRL-11268™) were transfected with 500 ng of RFX1 intron 7-luciferase reporter plasmid and 1 μg STAT3 expression plasmid or empty plasmid using Lipofectamine 2000. To detect the effect of IL-6 on the enhancer activity of intron 7, IL-6 (10 ng ml−1) was added into culture medium of CD4+ T cells after 24 h of transfection of the WT human RFX1 enhancer luciferase reporter construct. Each reporter experiment included 100 ng of renilla luciferase construct as an internal control. Forty-eight hours after transfection, cells were collected and lysed, and luciferase activity was quantified using the Promega Dual Luciferase Assay System (Promega) following the manufacturer's instructions.

**Electrophoretic mobility shift assays**. For EMSAs, two double-stranded DNA probes harboring two RFX1 binding sites of the human IL17A promoter were synthesized with biotin-labeled 3′ ends (Binding site 1: 5′- CAC-CAAGTTGCTTGGTAGCATGCAGGGTTGGAACATGCC-3′; Binding site 2: 5′-TTATATGATGGGAACTTGAGTAGTTTCCGGAATTGTCTCCACAA-CACCTGG-3′). Binding reactions were performed according to the manufacturer's instruction with a LightShift™ Chemiluminescent EMSA Kit (Thermo Scientific). Briefly, biotin end-labeled DNA containing the binding site was incubated with nuclear extract. This reaction was then subjected to gel electrophoresis on a native polyacrylamide gel and transferred to a nylon membrane. The biotin end-labeled DNA was detected using the Streptavidin-Horseradish Peroxidase Conjugate and the Chemiluminescent Substrate. For supershift assays, 2 μg of RFX1 antibody (Santa Cruz, Clone: I-19) was added to the binding reaction. Two hundred-fold molar excess of the cold unlabeled oligonucleotide was used for competition assays.

**Chromatin immunoprecipitation assays**. Anti-RFX1 antibody (Santa Cruz, Clone: I-19) used for ChIP assays was purchased from Santa Cruz. H3ac (cat. no. 39139), H3K4me3 (cat. no. 39915), and H3K9me3 (cat. no. 39765) antibodies were from Active Motif. Phosphorylated STAT3 antibody (clone: D3A7) was from CST. DNMT1 (cat. no. ab13537), HDAC1 (cat. no. ab7028), and SUV39H1 (cat. no. ab8898) were from Abcam. ChIP experiments were carried out with a ChIP Assay Kit (Millipore) according to the manufacturer's protocol. Briefly, 2 million cells were cross-linked with 1% formaldehyde, washed with cold phosphate-buffered saline, and lysed in buffer containing protease inhibitors (Roche). Cell lysates were sonicated to shear DNA and sedimented, and diluted supernatants were immunoprecipitated with the indicated antibodies. In all, 10% of the diluted supernatants were kept as "input" (input represents PCR amplification of the total sample). Real-time PCR primer sequences are shown in Supplementary Table 4. The amount of immunoprecipitated DNA was subtracted from the amount of amplified DNA, which was bound by the nonspecific normal IgG and subsequently calculated relative to the respective input DNA.

**BSP**. Genomic DNA was isolated from CD4+ T cells using the QIAamp DNA Mini Kit (Qiagen) according to the manufacturer's protocol. Bisulfite conversion was performed using the EZ DNA Methylation™ Kit (Zymo Research). The 264 bp fragment, including 7 CG pairs in the human IL17A promoter, the 421 bp fragment, including 7 CG pairs in the mouse Il17a promoter, and the 273 bp fragment, including 30 CG pairs in intron 7 of the RFX1 gene, were amplified by nested PCR and cloned into the pGEM-T vector (Promega). Ten independent clones were sequenced for each of the amplified fragments. Primers are listed in Supplementary Table 5.

**Generation of Rfx1f/f and Rfx1f/fCD4-cre mice**. All animal care protocols and experiments were reviewed and approved by the Animal Care and Use Committees of the Laboratory Animal Research Center at the Second Xiangya Medical School of Central South University. The Rfx1 locus (ENSMUSG00000031706, http://www.ensembl.org/index.html) is on chromosome 8 (Mus musculus), which includes 21

exons. To create loxP-*Rfx1*-loxP mice, a targeting vector was designed for insertion with a frt-flanked PGK-neo cassette and a loxP site upstream of exon 3 and a second loxP site downstream of exon 4. The loxP site is a 34 bp DNA sequence that can be recognized by Cre recombinase. If two loxP sites are introduced in the same orientation into a genomic locus, expression of Cre results in the deletion of the loxP-flanked DNA sequence. After linearization, the vector was electroporated into JM8A3 (C57BL/6N-derived embryonic stem (ES) cells). The collected ES cells were screened with 300 mg ml$^{-1}$ G418 and 2 mM Gan C for 8 days and assessed by PCR. The ES cells with correct homologous recombination were injected into blastocysts. After birth, the chimeric mice were bred with C57BL/6J mice to generate heterozygotes. At this point, the mutant mice were bred with Flp recombinase-expressing mice to remove the frt-flanked neo cassette. The resulting loxP-*Rfx1*-loxP mice were bred with CD4-cre transgenic mice to generate *Rfx1*$^{f/f}$CD4-cre mice. P1 and P2 were used to identify the genotypes of the *Rfx1* floxed allele (1058 bp) and the WT mice (842 bp). P1, 5′-AAGAGGCTCCTCAGACATAT-3′; P2, 5′-GTCTGGACCCACAGGCTTCC-3′. P3 and P4 were used to identify the genotypes of the cre allele (100 bp). P3, 5′-GCGGTCTGGCAGTAAAAACTATC-3′; P4, 5′-GTGAAACAGCATTGCTGTCACTT-3′. Mice were maintained in a specific pathogen-free animal facility. The loxP-*Rfx1*-loxP mice were generated by Shanghai Biomodel Organism Science & Technology Development Co. Ltd. CD4-cre mice (stock number: 022071) were purchased from The Jackson Laboratory (Bar Harbor, ME).

**EAE disease model**. EAE was induced by complete Freund's adjuvant (CFA)-MOG35-55 peptide immunization (China Peptides Biotechnology) and scored daily. Briefly, 8-week-old male mice were injected subcutaneously into the base of the tail and both flanks with a volume of 200 µl containing 200 µg MOG35-55 peptide emulsified in complete Freund's adjuvant (Sigma-Aldrich). Mice were also injected intraperitoneally with 500 ng of pertussis toxin (Sigma-Aldrich) on days 0 and 2 post-immunization. All reagents used for in vivo experiments were free of endotoxin. Mice were monitored daily for the development of disease, which was scored according to the following scale: 0, no symptoms; 1, tail weakness; 2, tail paralysis and hind limb weakness; 3, complete hind limb paralysis; 4, hind limb paralysis with forelimb weakness; and 5, moribund or death.

**MOG re-stimulation**. EAE was induced in *Rfx1*$^{f/f}$ mice and *Rfx1*$^{f/f}$-cre mice, and spleens were harvested on day 19. Isolated cells from the spleen were further cultured ex vivo with MOG35-55 (30 µg) for 3 days. IL-17 and IFN-γ concentrations in culture supernatant were measured by ELISA.

**Histology**. Spinal cords (4% paraformaldehyde fixed) were stained with H&E for the detection of inflammatory infiltrates and luxol fast blue for myelin detection. Histological sections were scored as follows: 0, no infiltration (<50 cells); 1, mild infiltration of nerve or nerve sheath (50-100 cells); 2, moderate infiltration (100-150 cells); 3, severe infiltration (150-200 cells); 4, massive infiltration (>200 cells). Paraffin-embedded sections of the kidneys were stained with H&E. To detect immune complex deposition, sections of frozen kidneys were stained for mouse C3 by indirect immunofluorescence with Rat anti-C3 antibody (abcam, clone: 11H9) and Cy3-conjugated goat anti-Rat IgG (Servicebio, cat. no. GB21302) and for mouse IgG by direct immunofluorescence with Alexa Fluor 488-conjugated goat anti-mouse IgG (abcam, cat. no. ab150113).

**Pristane-induced lupus mouse model**. Female Rfx1$^{f/f}$ and Rfx1$^{f/f}$-cre mice aged 8 weeks were injected intraperitoneally with 500 µl pristane (Sigma-Aldrich, USA). Urine samples were collected, and proteinuria were assessed using colormetric assay strip (URIT, China). The following scale was used for assessment: 0 = absent; ±=15 mg dl$^{-1}$; +=30 mg dl$^{-1}$; ++=100 mg dl$^{-1}$; +++=300 mg dl$^{-1}$; and ++++≥500 mg dl$^{-1}$. At 20 weeks, the end of observation period, serum was collected by cardiac puncture after anesthetizing mice. Serum anti-dsDNA IgG and IL-17 protein were determined by ELISA. Th1 and Th17 cells' proportions in splenic lymphocytes were detected using flow cytometry.

**Cytokine stimulation and inhibitor treatment**. Human total CD4$^+$ T cells were isolated from PBMCs of healthy individuals by CD4 magnetic beads. In all, $2 \times 10^6$ CD4$^+$ T cells were cultured in six-well plates and stimulated by TGF-β (5 ng ml$^{-1}$, PeproTech), IL-6 (10 ng ml$^{-1}$, PeproTech), IL-1β (10 ng ml$^{-1}$, PeproTech), or IL-23 (20 ng ml$^{-1}$, PeproTech), respectively. After 24 h, cells were collected for mRNA and protein detection. For inhibitor treatment experiment, $2 \times 10^6$ CD4$^+$ T cells were cultured in six-well plates and treated with STAT3 inhibitor S3I-201 (10 µmol l$^{-1}$) or NF-κB inhibitor QNZ (100 nmol l$^{-1}$) for 2 h, and then IL-6 (10 ng ml$^{-1}$) was added to stimulate T cells. After 22 h, cells were harvested for protein detection.

**Statistical analysis**. All data are presented as the mean ± s.d or s.e.m. Statistical analyses were performed in the GraphPad Prism version 5.0 software. Statistical significance was determined by *t*-tests (two-tailed) between two groups. When the sample data are not normally distributed, Mann–Whitney *U*-test (two-tailed) was used for statistical analysis. Two-way analysis of variance was used for comparisons of multiple groups. Sperman's correlation coefficient was used for the correlation

analysis. *P*-values < 0.05 were considered significant. No randomization or blinding was used, and no animals were excluded from analysis. Sample sizes were selected on the basis of previous experiments.

**Data availability**. The authors declare that the data supporting the findings of this study are available within the article and its supplementary information files or are available from the corresponding author upon request.

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

## Acknowledgements

This work was supported by the National Natural Science Foundation of China (No. 81522038, No. 81270024, and No. 81220108017), the Hunan Provincial Natural Science Foundation of China (14JJ1009), and the Project of Innovation-driven Plan of Central South University (No. 2016CX029).

## Author contributions

Q.L., M.Z., Y.T., Q.P., C.H., Y.G., and G.L. designed the experiments; M.Z., Y.T., Q.P., C. H., Y.G., G.L., B.Z., Y.H., and A.L. carried out most of the experiments; Z.W., M.L., and X.G. helped to collect the samples. R.W., H.W., and H.L. assisted with analysis and interpretation of experiments and results. Q.L. and M.Z. supervised the experiments, analyzed results, and wrote the manuscript.

## Additional information

**Competing interests:** The authors declare no competing financial interests.

