## [Peer Review File · Nature Communications]

Reviewers' comments:

Reviewer #1 (Remarks to the Author):

The authors show nice human experiments in which they demonstrate the link between RFX1 and IL-17. The present clear data on the epi control of IL17 by RFX1 and finally the control of RFX1 by STAT3. The mouse experiments provide final convincing data.

Some minor concerns:

1. They used CD4Cre PFX1 fl/fl mice, so they could not exclude perfectly the effect of other CD4+ subsets. They may want to consider some of the following experiments.

- a. To do a transfer model in EAE., and/or
- b. MOG restimulation study., and/or
- c. In vitro culture of other T cell subsets using PFX1 fl/fl mice.

2. In Fig. 1, they show that RFX1 negatively correlates with IL-17A level, but not with IL-17F. In Fig 6b, RFX1-deficient mice have more IL-17F expression than sufficient mice (although it is only on day7).The authors need to explain this discrepancy.

Reviewer #2 (Remarks to the Author):

In this very interesting and thorough work, Zhao et al present evidence for the role of RFX1 in the regulation of IL-17A production. The initial observation that led to this paper was that CD4+ T cells from SLE patients have decreased levels of this transcription factor. Correlative measurements using SLE T cells show that IL-17A mRNA and RFX1 mRNA may correlate. In downregulation/overexpression experiments the authors show that RFX1 may indeed directly regulate IL-17A production in both healthy and SLE CD4+ T cells. They constructed a conditional knockout animal that although phenotypically not abnormal, after being challenged with MOG, developed a worse phenotype than control mice. Finally the authors uncover a link between STAT3 activation following IL-6 (but interestingly not IL-23) and RFX1 downregulation. Overall this paper shows evidence for a transcription factor that may keep the IL-17A locus closed and upon release from the IL-17A promoter, it boosts IL-17A production. The main strong point of this paper is the thorough molecular biology work that links RFX1 to IL-17A. The main drawback is the limited evidence that this mechanism plays an important role in autoimmune diseases. The authors impetus for doing this work is based on their lupus work but they do not use a lupus model of disease to prove causation between low RFX1 in CD4 T cells and disease induced damage. The results using the EAE model are not as robust and hence no firm conclusion can be made on the importance of this pathway.

Specific comments

Fig 1. This is largely repetitive work (1a and b). In d statistical significance is based on outliers (active disease).

Fig 2. The siRNA experiments are interesting but seem to be done in cells that were not differentiated. How did IL-17A get transcribed in the absence of activation of relevant activating factors such as ROR γ t?

Fig 4. Does RFX1 directly bind to DNMTs or HDAC?

Fig 5. The differences in EAE severity are mild and possibly non statistically significant.

Fig 7-8. The link to IL-6 is interesting but SLE patients may not have upregulated IL-6 as CRP (a surrogate of IL-16) is notoriously normal in most SLE active patients. CRP and possibly IL-6 may be elevated in certain subcategories of SLE patients such as patients with active arthritis and serositis. It would be interesting to look at RFX1 in those patients in particular and correlate CRP levels in serum to RFX1 levels.

Reviewer #3 (Remarks to the Author):

This manuscript describes elegant work by Zhao and Lu and colleagues, addressing important and novel aspects of the epigenetic regulation of IL-17 in the context of autoimmunity. They report reduced expression of RFX1 in lupus CD4+ T cells, confirming their earlier work, and show that RFX1 downregulates IL-17 and Th17 expression and differentiation, via altering the epigenetic code in the IL-17 locus. They mechanistically link IL-6/STAT3 signaling to down-regulation of RFX1 using a series of nicely performed and convincing experiments. The data were supported by well designed experiments using in vitro and in vivo approaches, including a newly created RFX1 conditional deletion model and in EAE mouse model.

The paper is clearly written and I have no suggestions for modifications or improvement.

Point-by-Point Answers to the Reviewers' Comments

Reviewer #1 (Remarks to the Author):

The authors show nice human experiments in which they demonstrate the link between RFX1 and IL-17. The present clear data on the epi control of IL17 by RFX1 and finally the control of RFX1 by STAT3. The mouse experiments provide final convincing data.

Some minor concerns:

1. They used CD4Cre RFX1 fl/fl mice, so they could not exclude perfectly the effect of other CD4+ subsets. They may want to consider some of the following experiments. a. To do a transfer model in EAE, and/or b. MOG re-stimulation study, and/or c. In vitro culture of other T cell subsets using RFX1 fl/fl mice.

Response: We are very grateful for this critical comment. As suggested, we have performed MOG re-stimulation experiment. EAE was induced in *Rfx1^{fl/fl}* mice and *Rfx1^{fl/fl}-cre* mice, and spleen were harvested on day 19. Isolated splenocytes from the spleen were further cultured ex vivo with MOG (30µg) for 3 days. IL-17 and IFN-γ concentrations in culture supernatant were measured by ELISA. As shown below and also in supplementary Fig. 6, IL-17 production was increased in *Rfx1^{fl/fl}-cre* mice compared with *Rfx1^{fl/fl}* mice, whereas IFN-γ production did not reach a significant difference. The result was added in the revised manuscript (Page 6).

2. In Fig. 1, they show that RFX1 negatively correlates with IL-17A level, but not with IL-17F. In Fig 6b, RFX1-deficient mice have more IL-17F expression than sufficient mice (although it is only on day7). The authors need to explain this discrepancy.

Response: We thank the reviewer for this important comment. Although both IL-17A and IL-17F are primarily produced by Th17 lymphocytes, the regulatory mechanisms of them are different.

For example, according to the previous reports (Hedrich, et al. J Biol Chem. 2012 Feb 10;287(7):4715-25.), the expression of IL-17F was reduced in SLE T cells, which was suppressed by the increased transcription factor cAMP-responsive element modulator α (CREM α). In contrast, the other report (Ref.15) showed that CREM α can induce IL-17A expression in SLE T cells through altering epigenetic modifications in the promoter of *IL17A* gene. In our study, we found no significant difference in IL17F expression between SLE patient and healthy controls, even a somewhat small reduction of IL17F expression in SLE CD4⁺ T cells compared with healthy controls. There is no correlation between the reduced RFX1 expression and IL17F levels in CD4⁺ T cells of SLE patients. In Fig. 6b, we showed the increased IL17F expression in the induced Th17 cells with Rfx1 deficiency (derived from *Rfx1*^{fl/fl}-cre mice) compared with the cells with Rfx1 sufficiency (derived from *Rfx1*^{fl/fl} mice) in Th17-polarizing conditions. Our results showed that *Il17a* and *Il17f* expression, as well as the other two Th17 cell related genes *Il23r* and *Rorc*, were upregulated in the induced Th17 cells with Rfx1 deficiency compared with the cells with Rfx1 sufficiency, which indicated that Rfx1 deficiency promote Th17 cell differentiation in Th17-polarizing conditions. Taken together, the discrepancy may be due to two reasons as following: (1) the different samples were used to detect IL17F expression in the two experiments. The total CD4⁺ T cells of SLE patients and healthy controls were used in Fig. 1 and the induced Th17 cells from *Rfx1*^{fl/fl} mice and *Rfx1*^{fl/fl}-cre mice were used in Fig. 6b. (2) the complicatedly regulatory mechanism of IL17F expression in CD4⁺ T cells of SLE patients, which is different from the mechanism under Th17-polarizing conditions. This explanation has been added in the discussion section (Page 9).

Reviewer #2 (Remarks to the Author):

In this very interesting and thorough work, Zhao et al present evidence for the role of RFX1 in the regulation of IL-17A production. The initial observation that led to this paper was that CD4⁺ T cells from SLE patients have decreased levels of this transcription factor. Correlative measurements using SLE T cells show that IL-17A mRNA and RFX1 mRNA may correlate. In downregulation/overexpression experiments the authors show that RFX1 may indeed directly regulate IL-17A production in both healthy and SLE CD4⁺ T cells. They constructed a conditional knockout animal that although phenotypically not abnormal, after being challenged with MOG,

developed a worse phenotype than control mice. Finally the authors uncover a link between STAT3 activation following IL-6 (but interestingly not IL-23) and RFX1 downregulation.

Overall this paper shows evidence for a transcription factor that may keep the IL-17A locus closed and upon release from the IL-17A promoter, it boosts IL-17A production. The main strong point of this paper is the thorough molecular biology work that links RFX1 to IL-17A. The main drawback is the limited evidence that this mechanism plays an important role in autoimmune diseases. The authors' impetus for doing this work is based on their lupus work but they do not use a lupus model of disease to prove causation between low RFX1 in CD4 T cells and disease-induced damage. The results using the EAE model are not as robust and hence no firm conclusion can be made on the importance of this pathway.

Response: We agree with the reviewer and appreciate the reviewer for this insightful comment. Our work is mainly based on samples from lupus patients. The more strong evidence will be provided to prove causation between RFX1 downregulation in CD4⁺ T cells and lupus-induced damage if we can get similar results from a lupus model. In fact, in order to confirm this observation in a lupus model, we began to induce the lupus-like mouse model by intraperitoneal injection of pristane in *Rfx1*^{fl/fl}-cre mice (Rfx1 deficiency) and *Rfx1*^{fl/fl} mice (normal control) five months ago, because it will take a long time to induce a lupus-like mouse model by intraperitoneal injection of pristane. We have detected IL-17 levels and the changes of lupus characteristics in two groups of mice at 20 weeks after pristane injection. As shown below, the urine protein levels (**a**) and the serum levels of anti-dsDNA antibody (**e**) and IL-17 protein (**f**) were higher in *Rfx1*^{fl/fl}-cre mice compared to those in *Rfx1*^{fl/fl} mice ($P < 0.05$). H&E staining of kidney sections revealed more severe renal damage in *Rfx1*^{fl/fl}-cre mice than that in *Rfx1*^{fl/fl} mice (**b**). Immunostaining analysis revealed more deposition of IgG (**c**) and complement 3 (C3) (**d**) in glomeruli of *Rfx1*^{fl/fl}-cre mice compared with that in *Rfx1*^{fl/fl} control mice. The flow cytometry showed that the percentage of IL-17⁺CD4⁺ T cells in spleen was increased in *Rfx1*^{fl/fl}-cre mice compared with *Rfx1*^{fl/fl} mice (**g, h**). Together, these data indicate that Rfx1 deficiency promotes lupus autoimmunity and exacerbates renal damage, which suggests an important role of Rfx1 in the pathogenesis of SLE. These results and the method of pristane-induced mouse model were shown in the revised manuscript (Page 7 and Page 15) and Fig. 7.

Specific comments:

1. Fig 1. This is largely repetitive work (1a and b). In d statistical significance is based on outliers (active disease).

Response: We thank the reviewer for his/her comment. Fig 1a shows the mRNA expression levels of RFX1 and Fig 1b shows the protein levels of RFX1 in CD4⁺ T cells of SLE patients and normal controls. Both of them indicate that RFX1 expression was down-regulated in both mRNA and protein levels in CD4⁺ T cells of SLE patients compared with normal controls. We agree with the reviewer's comment about the statistical significance in Fig. 1d. To exclude the statistical error derived from some outliers in active SLE patients, we added the samples of patients with active SLE and normal controls, and detected the expression of IL17A and RFX1. We re-compared the mRNA expression of IL17A between active SLE patients (n=23) and normal controls (n=14) using Mann-Whitney U test. As shown below, IL17A mRNA expression was significantly increased in CD4⁺ T cells of active SLE patients compared with normal controls (P-value<0.01). The result is also shown in the revised Fig. 1.

2. Fig 2. The siRNA experiments are interesting but seem to be done in cells that were not differentiated. How did IL-17A get transcribed in the absence of activation of relevant activating factors such as ROR γ t?

Response: We thank the reviewer for raising this point and giving us this opportunity to make a clarification. Previously, several groups have identified some transcription factors such as Evt5 and CREM α to activate *IL17A* transcription through binding in the promoter region of *IL17A* gene via recruiting some epigenetic modifying enzymes, which is independent of ROR γ t (Ref. 14 and Ref. 15). In this study, we found that RFX1 can bind in the promoter region of *IL17A* and recruit DNMT1, HDAC1 and SUV39H1 to repress *IL17A* transcription directly through increasing DNA methylation and repressive histone marker H3K9 tri-methylation and decreasing active histone marker H3 acetylation. Acetylation of histone H3 contributes to transcriptional activation whereas DNA methylation and trimethylation of H3K9 repress gene transcription. DNA methylation and H3K9 tri-methylation were decreased and H3 acetylation was increased in the *IL17A* loci when RFX1 expression was knockdown by siRNA, which increased the accessibility in the promoter region of *IL17A* gene, resulting in the transcription activation of *IL17A*. The regulatory mechanism has been described in the discussion section of the revised manuscript (Page 9).

3. Fig 4. Does RFX1 directly bind to DNMTs or HDAC?

Response: Yes, RFX1 directly binds to DNMT1 or HDAC. The co-immunoprecipitation (co-IP) experiments have confirmed that RFX1 can directly bind DNMT1, HDAC1 and SUV39H1 to form a protein complex in CD4⁺ T cells in our previous studies (Ref. 16 and Ref. 21), which have been clarified in the first paragraph of discussion section in the revised manuscript.

4. Fig 5. The differences in EAE severity are mild and possibly non statistically significant.

Response: We appreciate the reviewer for the comment. In this study, we used two way ANOVA to compare the clinical EAE scores across the whole observed period, we present the higher EAE scores in *Rfx1^{fl/fl}*-cre mice than that in *Rfx1^{fl/fl}* mice with a statistical significance. We also applied Mann-Whitney U test (two-tailed) to compare the difference between *Rfx1^{fl/fl}* mice and *Rfx1^{fl/fl}*-cre mice at each time point and found that the difference of clinical EAE score was significant between *Rfx1^{fl/fl}* mice and *Rfx1^{fl/fl}*-cre mice after 11 days except on day 27. We mentioned the statistical methods in the revised figure legend of Fig. 5.

5. Fig 7-8. The link to IL-6 is interesting but SLE patients may not have upregulated IL-6 as CRP (a surrogate of IL-6) is notoriously normal in most SLE active patients. CRP and possibly IL-6 may be elevated in certain subcategories of SLE patients such as patients with active arthritis and serositis. It would be interesting to look at RFX1 in those patients in particular and correlate CRP levels in serum to RFX1 levels.

Response: We thank the reviewer for the insightful suggestions. As suggested, we measured the CRP levels in serum and analyzed the correlation between RFX1 expression levels and CRP levels in 20 SLE patients with active arthritis. As shown below, RFX1 expression levels are negatively correlated with CRP levels in SLE patients with active arthritis ($r=-0.713$, $P<0.001$). The result is shown in the Supplementary Fig. 9 and also added it into the revised manuscript (Page 7).

Reviewer #3 (Remarks to the Author):

This manuscript describes elegant work by Zhao and Lu and colleagues, addressing important and novel aspects of the epigenetic regulation of IL-17 in the context of autoimmunity. They

report reduced expression of RFX1 in lupus CD4+ T cells, confirming their earlier work, and show that RFX1 downregulates IL-17 and Th17 expression and differentiation, via altering the epigenetic code in the IL-17 locus. They mechanistically link IL-6/STAT3 signaling to down-regulation of RFX1 using a series of nicely performed and convincing experiments. The data were supported by well designed experiments using in vitro and in vivo approaches, including a newly created RFX1 conditional deletion model and in EAE mouse model.

The paper is clearly written and I have no suggestions for modifications or improvement.

Response: We appreciate the reviewer's encouraging and positive comments.

REVIEWERS' COMMENTS:

Reviewer #1 (Remarks to the Author):

the authors have addressed carefully all raised concerns.

Reviewer #2 (Remarks to the Author):

The authors have performed additional experiments that have strengthened their manuscript. I have no further concerns.

REVIEWERS' COMMENTS:

Reviewer #1 (Remarks to the Author):

the authors have addressed carefully all raised concerns.

Reviewer #2 (Remarks to the Author):

The authors have performed additional experiments that have strengthened their manuscript. I have no further concerns.

We would like to thank the reviewers for their overall encouraging comments regarding our revised manuscript.

The following is a point-to-point response to the two reviewer's comments.

Reviewer #1 (Remarks to the Author):

The authors have addressed carefully all raised concerns.

Response: We are glad to know that our revised version addressed Reviewer #1's concerns.

Reviewer #2 (Remarks to the Author):

The authors have performed additional experiments that have strengthened their manuscript. I have no further concerns.

Response: We are glad to know that our revised version addressed Reviewer #2's concerns.